# Selection-driven trait loss in independently evolved cavefish populations

Rachel L. Moran[1,2] ✉, Emilie J. Richards[1], Claudia Patricia Ornelas-García[3], Joshua B. Gross[4], Alexandra Donny [1], Jonathan Wiese[1], Alex C. Keene [2], Johanna E. Kowalko[5], Nicolas Rohner [6,7] & Suzanne E. McGaugh[1]

Laboratory studies have demonstrated that a single phenotype can be produced by many different genotypes; however, in natural systems, it is frequently found that phenotypic convergence is due to parallel genetic changes. This suggests a substantial role for constraint and determinism in evolution and indicates that certain mutations are more likely to contribute to phenotypic evolution. Here we use whole genome resequencing in the Mexican tetra, *Astyanax mexicanus*, to investigate how selection has shaped the repeated evolution of both trait loss and enhancement across independent cavefish lineages. We show that selection on standing genetic variation and de novo mutations both contribute substantially to repeated adaptation. Our findings provide empirical support for the hypothesis that genes with larger mutational targets are more likely to be the substrate of repeated evolution and indicate that features of the cave environment may impact the rate at which mutations occur.

Identifying the mechanisms that drive adaptation is fundamental to understanding evolution. Systems where similar phenotypes have evolved repeatedly across independent lineages can be leveraged to uncover the genetic basis of adaptive traits and provide insight into the predictability of the evolutionary process. Repeatedly evolved traits are often assumed to be adaptive and the result of strong positive selection[1–3] (but see refs. 4–6). Recent advances in analytical tools provide unprecedented opportunity to compare genome-wide patterns of genetic variation across lineages that have independently evolved similar phenotypes in response to similar environmental pressures. These powerful approaches can be used to infer the genetic and mechanistic basis of adaptive trait evolution and to better understand the predictability of evolution in natural populations[7–9].

Identifying the molecular basis of repeated evolution has traditionally posed a major challenge[10–12]. Similar phenotypic changes can result from selection targeting different genes within the same regulatory network, independent mutations within the same gene, or identical substitutions within the same gene (e.g.,[13–17]). Adding further complexity, repeated evolution can occur through three main processes, including sorting of standing genetic variation that was present in the ancestral population, transporting adaptive alleles via gene flow among populations experiencing similar selective pressures, or de novo mutations occurring at the same locus[3,6,8,18]. The first two modes are forms of "allele reuse," also called "sorting," and although selection on these alleles may occur multiple times independently across replicate populations, the adaptive allele only arose once[3]. Only in the case of repeated de novo mutations does the allele arise multiple times independently. Distinguishing among these three modes has been difficult, especially given that the same substitution could potentially occur multiple times independently in different lineages. Accordingly, empirical tests investigating the mode of repeated evolution have been exceedingly rare, though such tests are needed to inform our understanding of both the predictability of evolution and the characteristics of genes that repeatedly facilitate adaptation[3].

[1]Department of Ecology, Evolution, and Behavior, University of Minnesota, Saint Paul, MN, USA. [2]Department of Biology, Texas A&M University, College Station, TX, USA. [3]Colección Nacional de Peces, Departamento de Zoología, Instituto de Biología, Universidad Nacional Autónoma de México, Tercer Circuito Exterior S/N. CP 04510, D. F. México, México City, México. [4]Department of Biological Sciences, University of Cincinnati, Cincinnati, OH, USA. [5]Department of Biological Sciences, Lehigh University, Bethlehem, PA, USA. [6]Stowers Institute for Medical Research, Kansas City, MO, USA. [7]Department of Molecular & Integrative Physiology, KU Medical Center, Kansas City, KS, USA. ✉e-mail: rlmoran@tamu.edu

Organisms inhabiting caves have evolved a variety of phenotypes that have long fascinated biologists[19,20]. Caves are extreme environments with no light and much lower nutrient availability and dissolved oxygen compared to surface habitats[21,22]. In response to environmental pressures, cave organisms have repeatedly evolved regressive traits (i.e., losses or reductions from the ancestral state), such as reduced pigmentation, reduced or absent eyes, reduced sleep duration, and disrupted circadian rhythms, as well as constructive traits, such as altered metabolism and enhanced non-visual sensory capabilities (e.g., number of neuromasts used for sensing movement)[23]. The mechanism underlying regressive trait evolution has long been a point of controversy[24–26], as traits can be rendered non-functional through drift or selection, depending on whether loss has a neutral or positive effect on fitness, respectively. The evolution of regressive traits in cave organisms has historically been attributed to neutral processes[27–29], most famously by Darwin[30] but a growing body of literature suggests that loss-of-function mutations are often adaptive (reviewed in ref. 31). However, determining the relative contributions of neutral processes, direct selection, or indirect selection (e.g., due to hitchhiking or pleiotropy) to the evolution of loss of function remains a challenge.

Among the more than 200 cave animals found throughout the world, the Mexican tetra (*Astyanax mexicanus*), has emerged as a leading model in a wide array of biological fields, including evolution, development, neuroscience, and human disease, and provides an excellent system for studying the molecular basis of repeated evolution[32]. This species comprises surface ecotypes and derived cave ecotypes that are interfertile (Fig. 1a). Cave ecotypes of *A. mexicanus* are currently found in at least 30 caves in central Mexico (Fig. 1b), and geographically distinct groups of caves appear to have been colonized by different ancestral surface lineages[33–35]. Thus, this system is unique in that it provides multiple levels of replication for studies of the molecular basis of repeated evolution, both between different cave systems and among populations within a cave system. Furthermore, multiple categories of regressive and constructive traits have evolved repeatedly across populations of *Astyanax* that inhabitant caves, allowing for investigations into the relative contribution of deterministic versus stochastic processes in repeated trait evolution and the underlying mechanism of repeated evolution (i.e., allele reuse/sorting versus convergence). Together, these attributes make *Astyanax* an ideal system to study the mechanisms underlying adaptation and the predictability of evolution.

Here, we leverage large-scale whole genome sequencing of nearly 250 *A. mexicanus* individuals combined with cutting-edge population genomic methodologies[7,8,36,37] to quantify selection across the genome in multiple cave and surface populations to investigate how often we might expect repeated evolution of the same phenotype to take repeated paths at the molecular level. We also ask whether any commonalities exist among genes that are the repeated target of evolution. By analyzing the most expansive sampling to date in this system, we show that (1) at least two independent origins of cave phenotypes have evolved from two separate surface lineages, (2) selection has played a central role in driving the evolution of both constructive and regressive traits, (3) strong selective pressures in the extreme cave environment caused rapid evolution across multiple traits simultaneously at approximately the same time as cave invasions were estimated to have occurred, (4) selection has targeted the same genes repeatedly across cavefish lineages and this has proceeded largely via selection on standing genetic variation and de novo mutations, and (5) genes evolving repeatedly across cave lineages are longer and, thus, have a greater mutational opportunity compared to genes across the rest of the genome, a pattern that is primarily driven by genes with independent mutations across lineages. Overall, our work presents strong evidence that repeated evolution of the canonical cavefish phenotypes was shaped by selection and that alleles associated with constructive and regressive cave-derived traits were swept to fixation rapidly and

nearly simultaneously after initial invasion of the cave environment. More broadly, our work provides insight into the factors contributing to the repeatability of evolution and whether features such as coding sequence length may predictably bias evolution via novel mutations in certain genes.

## Results

### Cavefish evolved at least two independent times

Inferring the number of independent origins of cave adaptation is necessary to identify the evolutionary forces driving repeated evolution in *Astyanax mexicanus*. To investigate population structure and conduct comprehensive phylogenomic tests for repeated evolution of cave adaptation, we analyzed whole genome sequences from a total of 248 *A. mexicanus* individuals across 18 cave and eight surface populations throughout the range of *A. mexicanus* in northeastern and central Mexico (Fig. 1b; also see Supplementary Note 1), as well as four outgroup individuals (two *A. nicaraguensis*, two *A. aeneus*; for coverage and read counts for each sample see Supplementary Data 1). This represents the most extensive genomic dataset in this species to date.

The findings of our phylogenetic analyses using multispecies coalescent-based and gene tree-based approaches were largely in agreement with one another (Fig. 1c, d, Supplementary Figs. 1–4) and suggested three independent origins of cave adaptation: (1) Guatemala region cave populations and Jalpan, Caballo Moro, Río Choy, and Mante surface populations (collectively referred to as the "Lineage 1") (Garduño-Sánchez et al., in review), (2) El Abra region cave populations and Rascón, Gallinas, and Peroles surface populations (collectively referred to as the "Lineage 2"), and (3) the Micos region Subterráneo cave population and Micos surface population. However, we found substantial support for an alternative hypothesis that ongoing gene flow between Subterráneo cavefish and the nearby Micos River surface fish has led to Subterráneo grouping phylogenetically with the surface populations rather than with the other cave populations (see Supplementary Note 2, Supplementary Table 1, Supplementary Figs. 5, 6). Thus, our data support that Subterráneo cave originated from a Lineage 2 surface population (referred to in previous publications as the "old" lineage) but this signal is mostly obscured through hybridization with the Lineage 1 surface fish populations presently found in the Micos region (referred to in previous publications as the "new" lineage). The divergence of the caves from their surface counterparts occurred at approximately the same time (161–190 k generations ago) for each lineage, and thus, neither is a "new" or "old" cave lineage[35]. We therefore refer to these as "Lineage 1" and "Lineage 2" here.

In summary, our phylogenetic reconstruction clearly demonstrates that at least two independent cave colonization events have occurred stemming from two distinct surface lineages, with the Guatemala region cavefish originating from the Lineage 1 surface fish and the El Abra and Micos region cavefish originating from the Lineage 2 surface fish (Fig. 1c, Supplementary Figs. 1–4). Our data also indicate subsequent gene flow between a Micos region cave and surface fish from the Lineage 1, explaining the phylogenetic placement of Subterráneo cave as sister to Micos surface fish (rather than other Lineage 2 cavefish populations) in the present and previous studies (Supplementary Figs. 5, 6). Together, these key findings lay a framework for using the *A. mexicanus* system to study the genetic basis of repeated evolution of adaptive traits.

### Selection shaped cave-derived regressive traits

The contributions of selection to regressive trait evolution has been a highly controversial topic in the field of evolutionary biology[25,38]. Non-functionalization can occur through drift if losing the trait has a neutral effect on fitness. Alternatively, non-functionalization can occur through selection if trait loss is adaptive. To test the role of selection in the evolution of both regressive and constructive cave-derived traits, we used diploS/HIC[36,37] to quantify selection across the entire genome

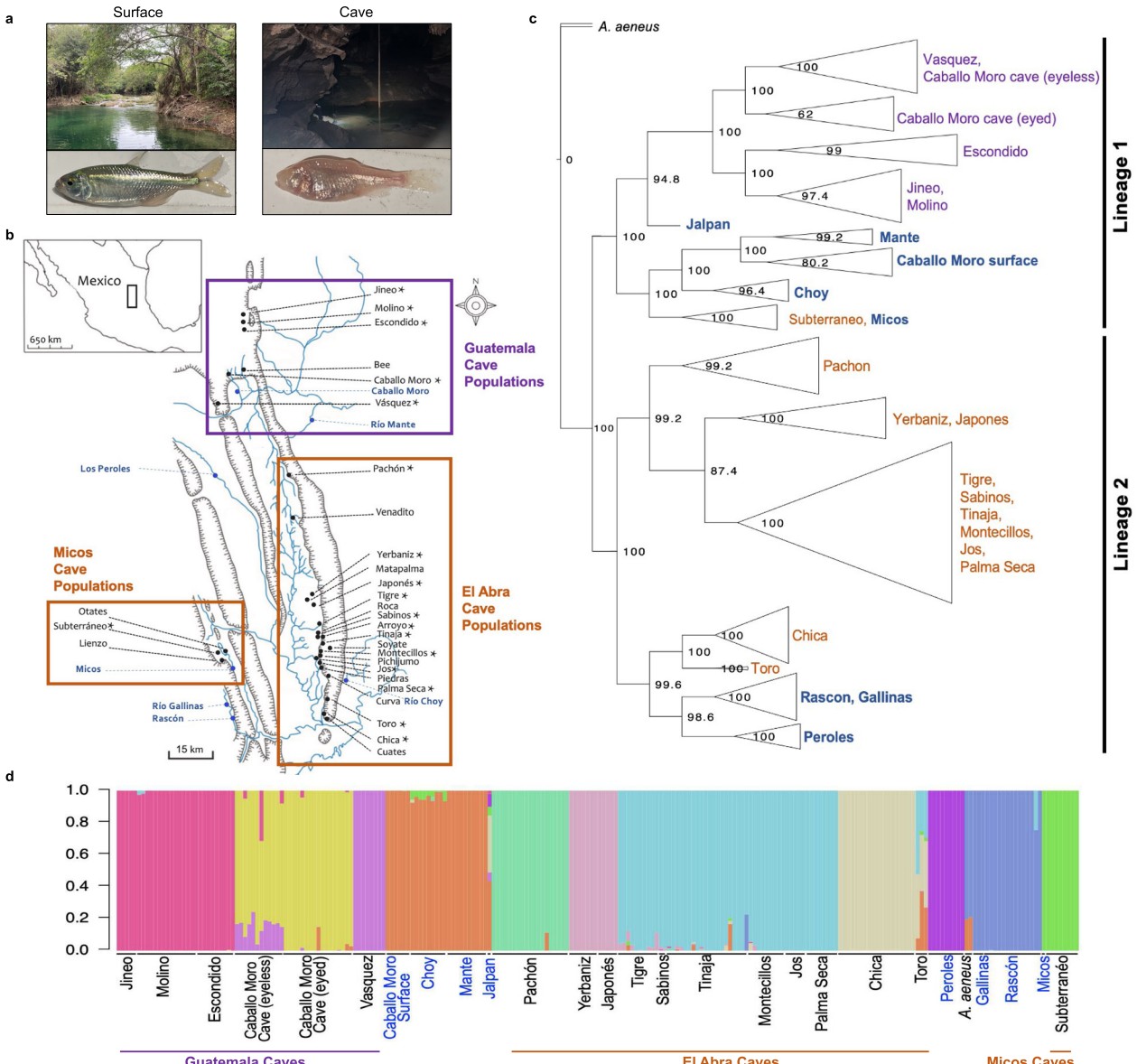

**Fig. 1 | Map of collections sites, phylogenetic tree, and population structure.** **a** Representative photos of a surface environment containing the surface *A. mexicanus* ecotype (left) and a cave environment containing the cave *A. mexicanus* ecotype (right). **b** Sampling locations for the present study. Locations of 29 caves within the Sierra de El Abra region of Mexico with known *Astyanax mexicanus* cavefish populations are labeled in black. Cavefish populations sampled for the current study (18 caves) are marked with an asterisk. The Guatemala caves are shown within a purple box. The El Abra and Micos caves are shown within orange boxes (these cave regions contain cavefish from a lineage of surface stock independent from the surface lineage that invaded the Guatemala caves). Surface fish populations sampled for the current study (8 surface locations) are labeled in blue. One surface population, Jalpan, is not shown (collected approximately 100 km south of the Subterráneo cave). Map modified with permission from Gross[50]. **c** Multi-species-coalescent tree inferred using 1,121,282 SNPs in SVDQuartets. SVDQuartets was run with a sampling of 500,000 random quartets and 500 standard bootstrap replicates specified to obtain bootstrap node support values shown. See Supplementary Fig. 2 for the tree with each sample labeled. **d** ADMIXTURE bar plot for K (number of unique genetic clusters) of 11. Surface populations are labeled in blue text and cave populations are labeled in black text. Source data are provided as a Source Data file.

in cavefish and surface fish from both lineages. For this analysis, we focused on seven cavefish populations and five surface fish populations with the highest sequencing coverage and sample size (see Supplementary Data 1 for coverage details; see Supplementary Table 2 for populations and sample sizes).

In each population, we used diploS/HIC to conduct scans for selection in 5 kb windows across the genome (Dryad repository, https://doi.org/10.5061/dryad.3xsj3txmf). We found evidence of selection in 1.5X more of the genome in cave populations compared to surface populations (Supplementary Fig. 7, Supplementary Table 3; mean ± SE proportion of 5 kb genomic windows under selection in surface populations = 0.143 ± 0.027; mean ± SE proportion of 5 kb

genomic windows under selection in cave populations = 0.213 ± 0.006; one-sided Wilcoxon rank sum test: W = 29, *p*-value = 0.037). This indicates that on average a larger proportion of cavefish genomes carry signatures of positive selection compared to surface fish genomes. We note that population bottlenecks can cause neutrally evolving genomic regions to exhibit low genetic variation that mimic a selective sweep. However, the demographic models used by diploS/HIC to detect regions under selection account for past population size changes and reduce the rate of false positives. Furthermore, the observed result that a larger proportion of the genome is under selection in cavefish compared to surface fish populations was robust to demographic model misspecification (i.e., when identical

demographic models were assigned to all populations, regardless of cave or surface identity, the results were qualitatively the same).

We identified genes that had evidence of a soft or hard selective sweep in each cave population and no sweeps in a same-lineage surface population. Notably, 30% of all genes under selection in caves were shared between lineages, suggesting the same genes are often used in repeated evolution in this system (Supplementary Fig. 8a; see below). This analysis revealed that between 1777 to 6238 genes per cave population were associated with hard or soft sweeps (Supplementary Tables 2, 3; Dryad repository, https://doi.org/10.5061/dryad.3xsj3txmf). On average, the number of genes with evidence of selection within each individual cave population was 3730, which represents 14% of all annotated genes in the genome (see Supplementary Table 2, Supplementary Fig. 8b, c, Dryad repository, https://doi.org/10.5061/dryad.3xsj3txmf) and is in line with the percentage of the genes in the genome under selection in domesticated lineages[39].

We refer to genes with evidence of a selective sweep in cave populations and neutral evolution in surface populations as cave-adaptive alleles. Genes with cave-adaptive alleles were enriched for functional categories that are typically associated with both regressive cave-derived traits (e.g., eye morphogenesis, retinal development, light absorption, response to light) as well as traits thought to enhance survival in the cave environment (e.g., lipid metabolism, response to insulin, otolith formation, lateral line nerve development, DNA repair) (Supplementary Data 2). By comparison, an average of 920 genes (3% of all annotated genes) exhibited a sweep within a single surface population and neutral evolution in same-lineage cave populations (Rascón: 739 genes, Mante: 1101 genes; Supplementary Data 3). GO terms that were consistently enriched in genes under selection in all seven cave populations but not enriched in genes under selection in either surface population included organ morphogenesis, circulatory and epithelium development, and cranial skeletal system development.

Together, our analyses show that (1) a larger proportion of the genome has evidence of selective sweeps in cave populations compared to surface populations (Supplementary Fig. 7), (2) there is ample evidence of selection on genes associated with cave-derived regressive traits (i.e., eye development, pigmentation) and constructive traits (i.e., metabolism, non-visual sensory system development) (Supplementary Data 2, Supplementary Figs. 9, 10), and (3) nearly one-third of cave-adaptive alleles are shared between lineages (Supplementary Figs. 8a, 9a), suggesting that the same genes are often the target of repeated evolution of traits associated with cave adaptation in this system.

## Variants in selective sweeps appeared simultaneously across classes of cave-derived traits

Here, we estimated the timing of cave-derived variants in selective sweeps for genes with GO terms related to multiple different categories of regressive and constructive cave-derived phenotypes (Supplementary Data 2, Supplementary Tables 4, 5; Supplementary Figs. 10, 11) using Genealogical Estimation of Variant Age (GEVA)[40]. Across the seven cave populations examined for sweeps, the average estimated age of derived variants in selective sweeps in genes associated with cave-derived traits obtained from GEVA ranged from 150–190 k generations ago, with an assumed generation time of 1 year (Fig. 2, Supplementary Fig. 12; Supplementary Data 4). This corresponds almost exactly to our estimates of when ancestral surface stocks first invaded caves, 160–190 k generations ago (Fig. 2a, Supplementary Note 3[35]). There were no statistically significant differences in the timing of derived variants in selective sweeps across individual phenotypic categories within any of the cave populations examined (Supplementary Table 6) and there was also no difference in derived variant timing between traits when grouped as regressive versus constructive traits (Supplementary Table 7). Instead, we observed tight distributions near

the time of cave-surface divergence for each of the phenotypic categories. This suggests that cavefish adapted quickly to the novel cave environment, with strong selective pressures likely driving concurrent changes across many traits. Previous work has provided evidence that Yerbaniz cave experiences ongoing gene flow with the local surface fish population (see ref. 41), which we hypothesize to be the cause of the younger estimated sweep times on average compared to the other cave populations.

## Identification of repeated molecular evolution across lineages

We identified extensive evidence of repeated evolution between and within cavefish lineages using two approaches (Fig. 3). First, we compared sweeps identified by diploS/HIC (as described above) to identify shared selective sweeps across lineages. In total, 3710 genes had a sweep in at least one Lineage 1 and one Lineage 2 cave population (and neutral evolution in surface populations; Supplementary Data 2, Dryad repository, https://doi.org/10.5061/dryad.3xsj3txmf). The approach we used to investigate the underlying mode of repeated evolution (see below) requires three or more populations experiencing selection at a given locus and is computationally intensive. With this in mind, we identified a subset of 760 genes exhibiting shared selective sweeps across three cave populations (two from Lineage 2 lineage: Pachón and Tinaja; one from the Lineage 1: Molino) and neutral evolution in surface populations from both lineages (Fig. 3a, Supplementary Data 5).

Second, we used AF-vapeR[7] to scan the genome for differences in allele frequencies between replicate surface and cave populations indicative of allele reuse (i.e., selection on the same allele in both lineages) and locus reuse (i.e., selection on different alleles but at the same locus in both lineages) (Fig. 3a) in 50 SNP windows (median physical window size of 7.6 Kbp). Rather than scanning individual cave populations for evidence of positive selection, this multivariate approach uses eigen decomposition to scan the genome for allele frequency changes indicative of repeated selection across multiple replicate population pairs. We identified 47 windows (overlapping 34 genes) that showed a signature of allele reuse (Fig. 3b, e, Supplementary Fig. 13, Supplementary Data 6), with the same non-surface alleles increasing in frequencies across all seven of the cave populations examined (i.e., evidence that the same mutation has been selected for repeatedly across lineages; Fig. 3d).

We also identified 20,151 windows (overlapping 3590 genes) with AF-vapeR that showed a signature of locus reuse in all seven cave populations (Fig. 3b, Supplementary Data 6), where allele frequency change has repeatedly targeted the same locus within each lineage but followed unique genetic trajectories between lineages (i.e., different alleles are selected for in each cave lineage; Fig. 3f). Notably and as a check for validity, the list of locus reuse genes included *oca2* (Fig. 3f), which is a pleiotropic gene involved in sleep and pigmentation in cavefish[42–44], and has unique exon deletions in Lineage 1 and Lineage 2 cavefish populations. Nine genes were found to overlap both allele reuse and locus reuse windows (Fig. 3b; Supplementary Data 6) with different parts of the gene classified as allele reuse and locus reuse, though most do not have functional annotation data. These genes likely overlapped both categories because they tended to be very long. Overall, these results suggest that most of the repeated evolution in this system evolves from selection on unique mutations at the same loci, as opposed to reuse of the exact same substitution in both lineages.

These two methods were largely discordant in identifying genes involved in repeated evolution. Of the 760 genes with overlapping sweeps between Pachón, Tinaja, and Molino identified by diploS/HIC, 150 overlapped with locus reuse genes (Fig. 3b), yet, none were identified as allele reuse genes. Four of the allele reuse candidate genes identified with the AF-vapeR scan were also classified by diploS/HIC as having a hard or soft sweep in at least one cave in both Lineage 1 and Lineage 2 caves and neutral evolution in surface populations from both

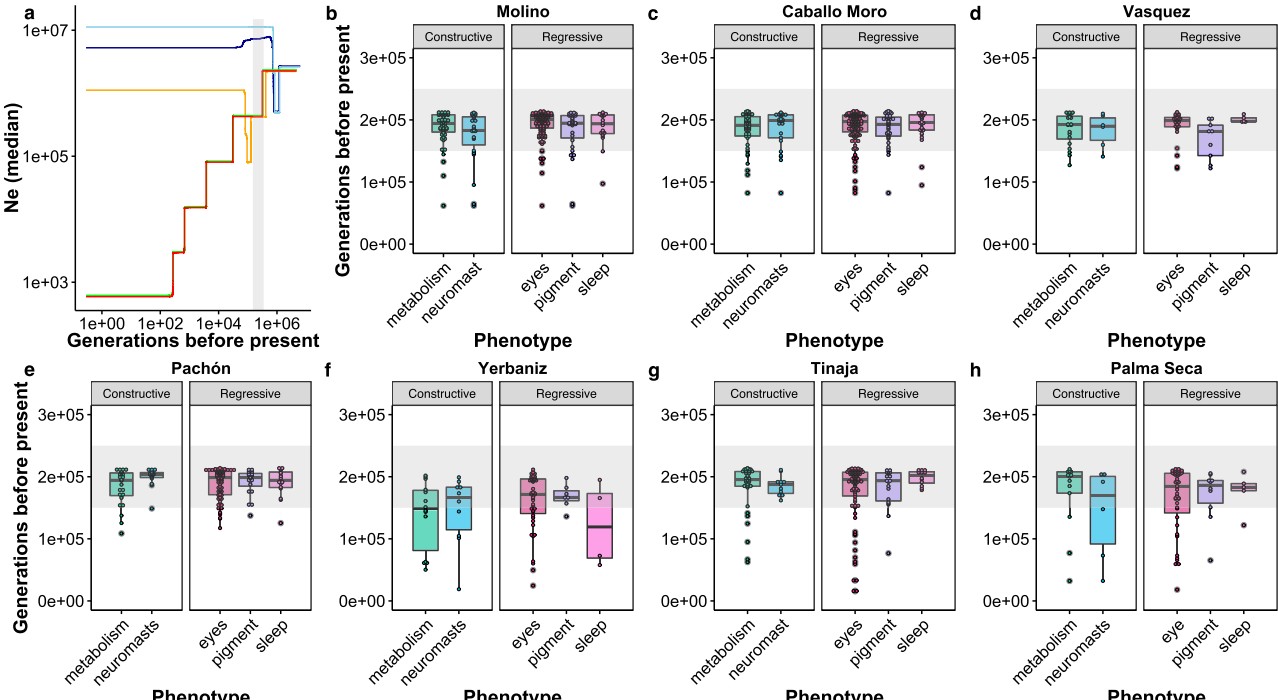

**Fig. 2 | Estimated demographic history and ages of variants in selective sweeps for cave populations. a** Stairway plot showing median Ne over time in Pachón (orange), Tinaja (red), and Molino (green) cave populations and Rascón (light blue) and Río Choy (dark blue) surface populations. Bottlenecks for present-day cave populations corresponding to the initial cave invasion by ancestral surface stock are highlighted by a gray rectangle (150,000–250,000 generations before present, spanning the range of previous demographic model-based median estimates for split times between cave and surface populations from each lineage from ref. 35). Note a more ancient bottleneck in the two surface populations shown (Rascón and Río Choy) around 800,000 generations before present, likely corresponding to migration into northern Mexico. **b–h** Ages of variants in selective sweeps with GO terms associated with constructive and regressive cave-derived phenotypes (see

Supplementary Data 4) in seven cave populations. For each box plot the horizonal line shows the median variant age, the shaded box spans the 25th to 75th percentile range, and the whiskers span the lowest to highest values that fall within 1.5 * the inter-quartile range. Raw data are shown over the box plots, with each dot representing a single gene. The number of independent biological replicates for each of the seven cave population included in **b–h** ranged from $n = 7$ to $n = 18$ (see Supplementary Table 3). See Supplementary Data 4 and Source Data for the number of genes in each phenotypic category within each population. Lineage 1 caves from the Guatemala regions are shown in **b–d**. Lineage 2 caves from the El Abra regions are shown in **e–h**. Gray rectangles span 150,000–250,000 generations before present. Source data are provided as a Source Data file.

lineages (out of 3710 genes total with sweeps shared across both cave lineages; Supplementary Fig. 8, Supplementary Data 2, 6). Furthermore, 549 of the 3590 locus reuse candidate genes identified with the AF-vapeR scan for repeated selection were also classified by diploS/HIC as having a hard or soft sweep in at least one Lineage 1 and one Lineage 2 cave and neutral evolution in surface populations from both lineages (out of 3710 genes total with sweeps shared across both cave lineages; Supplementary Fig. 8, Supplementary Data 2, 6). Notably, all of the candidate genes identified by AF-vapeR were classified as having a sweep or being linked to a sweep by diploS/HIC (Supplementary Data 6). We suspect that the discordance between the overlapping sweeps and AF-vapeR candidate gene sets may be because AF-vapeR considers phylogenetic context, and because many genes were excluded from the overlapping sweeps method if there was any evidence of sweeps in surface populations (a prerequisite imposed by the DMC analysis).

### Genes experiencing repeated evolution are linked to QTL and enriched for cave-derived phenotypes

The genes identified as experiencing some form of repeated evolution through selection are linked to both regressive and constructive cave-derived traits and to previously identified QTL associated with cave-derived phenotypes, which were identified mostly in Pachón x Lineage 1 surface hybrids. The genes showing a signature of allele reuse between lineages were not statistically significantly concentrated in one region, but four chromosomes (6, 12, 16, 22) contain nearly half of the 34 genes (Supplementary Data 5, 6), and 53% of these genes fell

within known QTL regions for cave-derived traits, including two regions which have been previously noted to harbor a high concentration of QTL (LG2 and LG17[35,45]; Fig. 3f, Supplementary Fig. 14, Supplementary Data 6). ShinyGO indicated there was a statistically significant clustering of genes with evidence of locus reuse across the genome. All enriched regions were connected to QTL, including the two regions with a high concentration of QTL (LG2 and LG17[35,45]; Fig. 3f, Supplementary Data 6).

Notably, we found that genes associated with previously identified eye size QTL (Dryad repository, https://doi.org/10.5061/dryad.3xsj3txmf) are enriched in the set of genes under selection within individual cave populations and in the set of genes that were identified as candidates for repeated evolution across cave lineages. A total of 6223 out of 26,698 (23%) genes in the surface fish genome annotation are associated with previously identified eye size QTL (Dryad repository, https://doi.org/10.5061/dryad.3xsj3txmf). Compared to all genes in the genome, those identified as being under selection within any of the seven cave populations analyzed (Supplementary Data 2, Supplementary Fig. 8) were enriched for eye size QTL (3272 out of 12,389 genes under selection only in cave populations; Fisher's Exact Test, $P < 0.0001$; Supplementary Data 5). Candidate gene sets for repeated selection were also enriched for eye size QTL. In the overlapping selective sweeps candidate gene set (genes with selective sweeps in Pachón, Molino, and Tinaja, neutral evolution in surface populations), 203 out of 760 (27%) of genes were associated with eye size QTL. This is a significantly higher proportion of genes than expected by chance (Fisher's Exact Test, $P < 0.01$; Supplementary Data 5). In the AF-vapeR

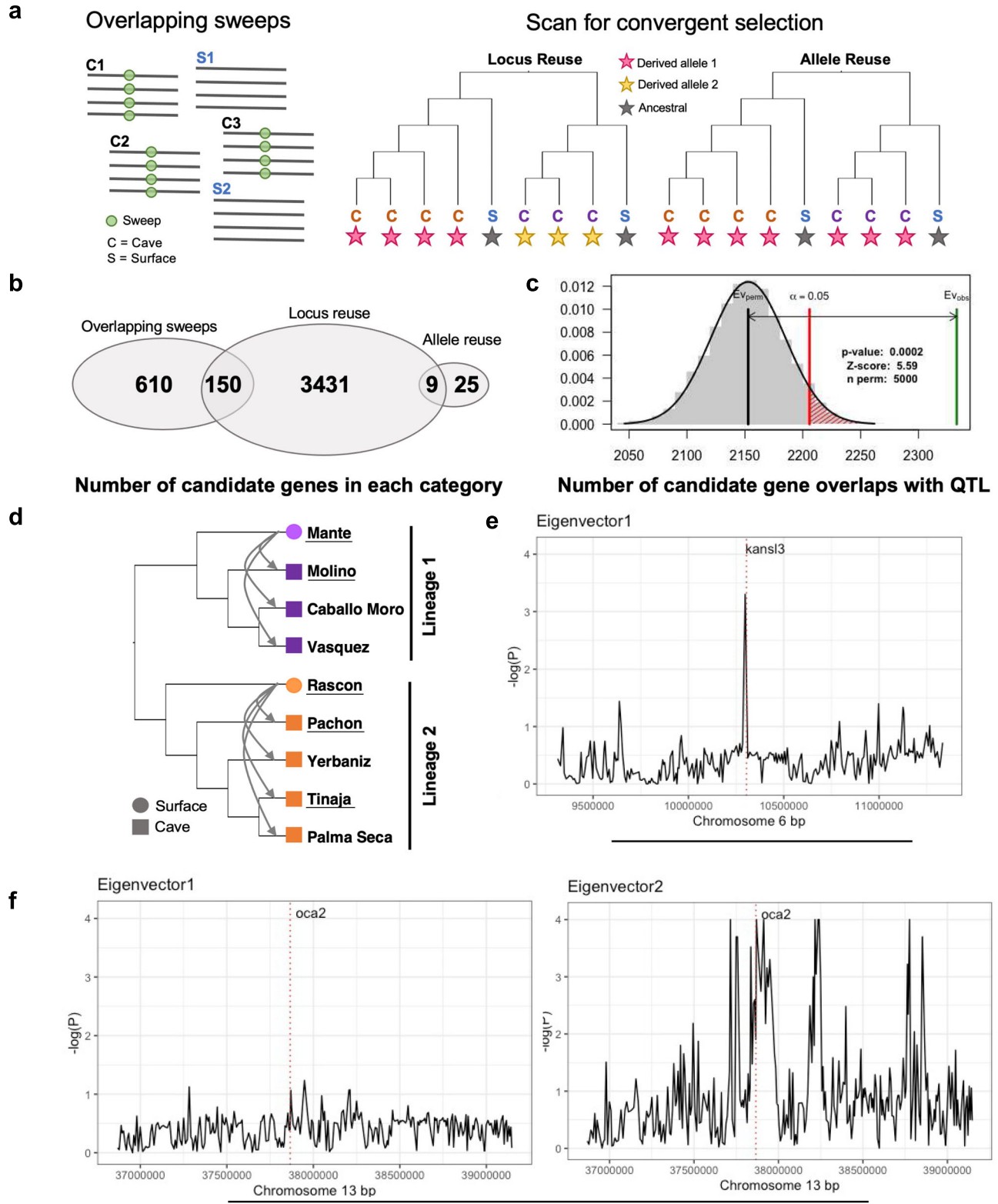

locus reuse candidate gene set (genes with evidence of repeated allele frequency change across all seven cave populations), we also saw enrichment of genes associated with eye size QTL (including *oca2*, *cbsa*, and *shha*; Supplementary Data 25). Specifically, 35% of locus reuse genes were associated with eye size QTL, significantly higher than what would be expected by chance (Fisher's Exact Tests, $P < 0.0001$; Supplementary Data 5). This further supports our finding that regressive traits are under selection in caves.

Of the 4225 total genes identified as being the repeated target of selection across cave lineages (Fig. 3b), 4066 occurred on an assembled chromosome and 2431 (60%) of these overlapped with previously identified QTL regions associated with cave-derived traits, including eye size, melanophore count, feeding behavior, body weight, and activity levels (Fig. 3c, Supplementary Fig. 14, Supplementary Data 5, 6). This amount of overlap was higher than expected by chance (permutation test, Z-score: 5.59, $P < 0.001$, based on 5000 permutations).

**Fig. 3 | Identification of candidate genes for repeated evolution between cavefish lineages. a** We took two approaches to identify loci evolving repeatedly across cavefish lineages. First, we scanned the genome in 5 kb windows and identified loci with overlapping selective sweeps in three cave populations and neutral evolution in two surface populations. Gray lines represent alleles within a population and green circles represent the location of a sweep. Second, we conducted scans for repeated evolution using AF-vapeR. This allowed us to detect patterns of allele reuse (i.e., selection at the same locus for the same cave-derived allele in both lineages; pink star represents selected derived allele in cave populations) or locus reuse (i.e., selection at the same locus but on unique alleles between lineages; pink and yellow stars represent two unique selected derived variants in cave populations). **b** The number of candidate genes identified in each category depicted in **a**. Note that some larger genes overlapped both allele reuse and locus reuse genomic windows ($n = 9$). **c** Predicted (black line) and observed (green line) overlap between

all candidate genes for repeated evolution across the 25 assembled chromosomes ($n = 4085$ genes total) and previously identified QTL regions for cave-derived traits (one-sided permutation test based on 5000 permutations, Z score = 5.59, $P < 0.001$). **d** Tree for populations included in the AF-vapeR analysis. Underlined populations (two surface and three cave) were used in the overlapping sweeps analysis. Replicate surface-cave comparisons across the Lineage 1 ($n = 3$) and Lineage 2 ($n = 4$) are indicated with arrows. **e** Repeated evolution via reuse of the same allele across the seven cave populations examined is seen as a significant loading (peak) on eigenvector 1. Here such a pattern is shown on chromosome 6, overlapping the location of *kansl3*. **f** An example of repeated evolution via locus reuse (i.e., selection on the same locus follows multiple trajectories, with different alleles under selection in different lineages) is seen as a significant loading (peaks) on eigenvector 2. Here such a region is highlighted on chromosome 13, overlapping the location of *oca2*. Source data are provided as a Source Data file.

Conversely, 72 out of 172 (42%) of genes evolving repeatedly across surface lineages fell within QTL regions, which does not deviate from what would be expected by chance ($X_1^2 = 1.51$, $P = 0.22$). This strongly suggests that the genes identified as under selection in multiple cave populations play a role in known cave-evolved traits.

The candidate genes for repeated evolution among cave populations were also enriched for GO terms associated with known cave-adaptive phenotypes. The 3590 genes with signatures of repeated evolution via locus reuse were enriched for ontologies related to cave-derived traits, including circadian rhythm (e.g., *per2*, *per3*, *cry1a*, *rora*, *rorc*), pigment development (e.g., *oca2*, *igsf11*, *ednrba*, *pax7b*), eye morphogenesis (e.g., *otx2*, *foxc1b*, *pitx2*), and metabolism (e.g., *pdx1*, *irs2a*, *irs2b*, *irs4a*) (Supplementary Data 7). Likewise, the set of 760 genes with overlapping sweeps in Pachón, Tinaja, and Molino cave populations were also significantly enriched for ontologies related to traits known to play key roles in cave adaptation, including visual perception and response to light stimulus, melanosome transport and localization, response to starvation and insulin, response to pH, and locomotory behavior (Supplementary Data 7). The list of 34 genes showing repeated evolution via allele reuse across cave populations from both lineages were enriched for ontologies related to iron-sulfur cluster binding, DNA polymerase activity, the DNA biosynthetic process, and catalytic activity acting on DNA (Supplementary Data 7). However, we note that because there were only 34 candidate genes in this category, and many were novel genes with no functional annotation, GO enrichment analysis may be underpowered for the allele reuse genes. In contrast, the set of 172 genes under selection in surface populations and with neutral evolution in cave populations did not exhibit enrichment for known cave-adaptive phenotypes (Supplementary Data 3, 7). Instead, we observed a significant enrichment of ontologies related to broad functional categories, including regulation of gene expression, transcription, and meiosis. Overall, these analyses support the conclusion that cave-derived traits, even regressive ones, are shaped extensively by natural selection on a subset of genes that are the repeated target of selection in both lineages.

### Standing genetic variation and de novo mutation drive repeated evolution across lineages

Whether repeated evolution proceeds from completely independent mutations or from repeated selection on the same pool of standing genetic variation is one of the most compelling questions in evolutionary biology. We determined whether repeated evolution proceeded through selection on standing genetic variation, migration, or de novo mutations through a powerful, recently developed method for distinguishing among modes of repeated evolution, "Distinguishing Modes of Convergence" (DMC)[8,46]. Because this analysis is computationally intensive, we focus on the 34 genes identified under repeated selection via allele reuse across seven cave populations by AF-vapeR and the 760 genes with overlapping selective sweeps across three cave populations identified by diploS/HIC (the latter of which included 150 genes with strong evidence of locus reuse across lineages

of the 3590 genes identified as evolving repeatedly via locus reuse across cave lineages by AF-vapeR) (Fig. 3b). One of the 34 genes identified by the AF-vapeR analysis could not be analyzed in DMC due to computational limitations imposed by its large size (>400 kbps; see Supplementary Data 5). Thus, a total of 793 genes were analyzed by DMC.

Repeated evolution through allele reuse predicts that the same alleles are under selection in both lineages, a pattern most likely to arise due to sorting of ancestral variation or gene flow between lineages. As expected, for the 33 genes identified as evolving repeatedly via allele reuse across lineages, DMC analyses suggested that selection on standing genetic variation was implicated as the primary mode of repeated evolution (25 out of 33 genes, 76%; Supplementary Data 5). Support for repeated evolution through selection on de novo mutations and migration between caves across lineages were observed in 18% (6 out of 33) and 6% (2 out of 33) of these genes, respectively (Supplementary Data 5).

For the 760 overlapping sweeps genes (which contained 150 genes with strong evidence of locus reuse across lineages), DMC again indicated support for repeated evolution via selection on standing genetic variation across all three cave populations in most genes (404 out of 760; 53%) (Fig. 4a; Supplementary Data 5). Strikingly, a model of repeated evolution via independent mutations among the two lineages was implicated in 40% of these candidate genes (304 out of 760; Supplementary Data 5). A model of repeated evolution via migration among cave populations between lineages was supported in only 7% (52 out of 760; Supplementary Data 5) candidate genes, indicating that adaptation via migration between Lineage 1 and Lineage 2 cave populations may be rare.

The percentages across different modes of repeated evolution as estimated in DMC in these gene sets (i.e., overlapping sweeps and allele reuse) are notably different. The 33 genes evolving repeatedly via allele reuse across lineages show a greater proportion of evolution from standing genetic variation (76%), as is expected since the exact same alleles are the target of selection in each population, whereas the 760 genes may be more reflective of an unbiased set of genes under selection repeatedly (53% evolution from standing genetic variation).

Repeated evolution through locus reuse predicts that unique alleles are under selection in each lineage at the same locus. Indeed, DMC indicated that compared to the entire set of 760 overlapping sweep genes analyzed, we observed that the 150 genes within this group that showed evidence of locus reuse (from the AF-vapeR analysis) showed a higher than expected number of genes evolving repeatedly through de novo mutations (52%; 78 out of 150) when compared to the entire 760 gene set (40% from de novo mutations) (Fisher's exact test: $P < 0.001$; Supplementary Data 5).

As expected, across the 793 total genes where the mode of repeated evolution was investigated with DMC, results indicated that the time the beneficial variant was present and segregating in the population was estimated to be much smaller for genes that

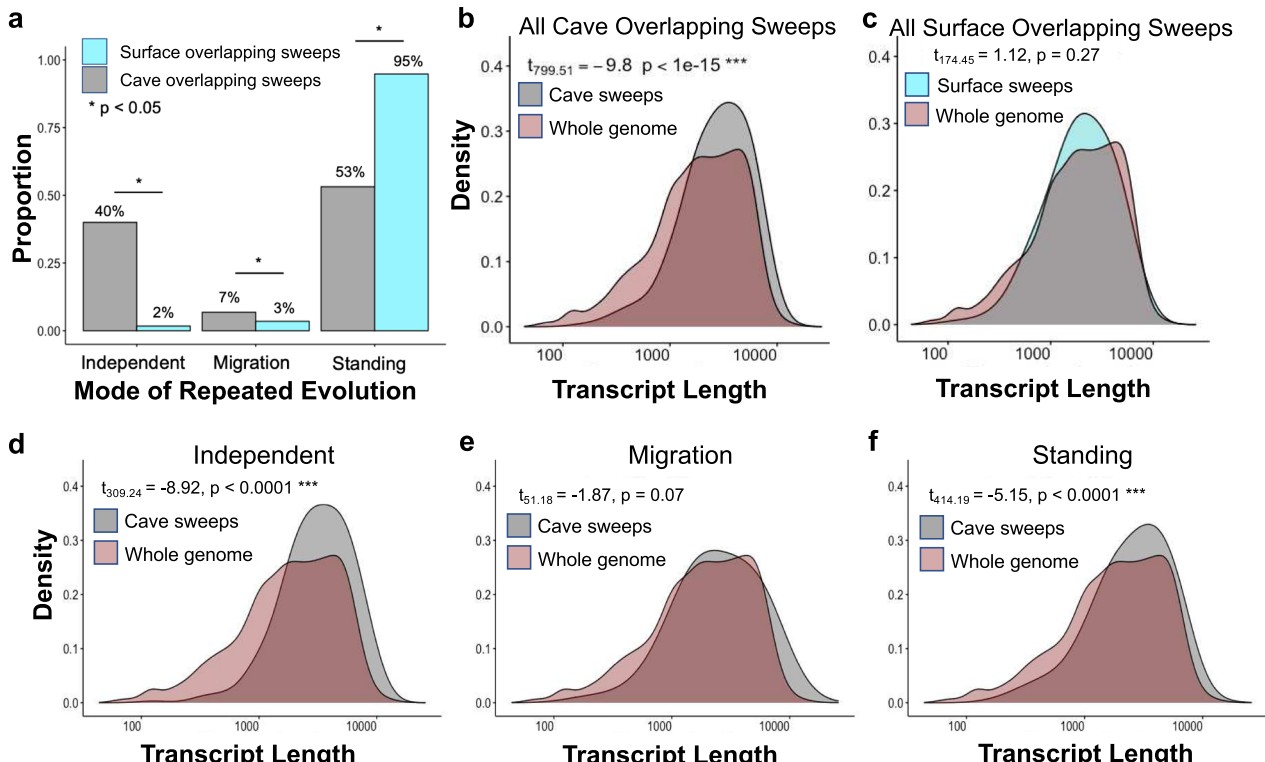

**Fig. 4 | Mode of repeated evolution for shared sweeps in cave and surface populations. a** Proportion of the 760 genes with shared sweeps across cave populations (gray) and 172 genes with shared sweeps across surface populations (blue) assigned best fitting models of repeated evolution via independent mutations, migration, or standing genetic variation in DMC. *$P < 0.05$ (Fisher's exact tests). **b** Density plot for transcript length (including UTRs and CDS) for all overlapping cave sweeps (gray) compared to the whole genome (red) (two-sided *t*-test, $t = -9.80$, df = 799.51, $P < 2.2e-16$). **c** Density plot for transcript length (including UTRs and CDS) for all overlapping surface sweeps (blue) compared to the whole genome (red) (two-sided *t*-test, $t = 1.12$, df = 174.45, $P = 0.27$). **d–f** Transcript length density plots for overlapping cave sweeps compared to the whole genome, broken down by predicted mode of repeated evolution: independent mutations (**d**; two-sided *t*-test, $t = -8.92$, df = 309.24, $P < 2.2e-16$), migration (**e**; two-sided *t*-test, $t = -1.87$, df = 51.18, $P = 0.07$), and standing genetic variation (**f**; $t = -5.15$, df = 414.99, $P = 4.03e-07$). ****P < 0.00001$. Source data are provided as a Source Data file.

experienced repeated selection via independent mutations compared to selection on standing genetic variation (mean ± SE time standing, independent model: 5.00 ± 0.00 generations; mean ± SE time standing, standing variation model: 26.99 ± 1.25 generations). As a secondary check, we used $D_{XY}$ to estimate the time to the most recent common ancestor between cave and surface alleles (Supplementary Data 5). Consistent with the expectations from mode of repeated evolution estimated by DMC, genes evolving via standing genetic variation had older split times between cave and surface populations (mean ± SE generations before present: Molino-Mante split = 324,075.65 ± 20,268.07; Pachón-Rascón split = 282,110.12 ± 22,743.69; Tinaja-Rascón split = 261,587.20 ± 21,990.97) compared to those evolving repeatedly via migration and independent mutations (mean ± SE generations before present: Molino-Mante split = 244,061.53 ± 14,230.83; Pachón-Rascón split = 214,818.03 ± 17216.96; Tinaja-Rascón split = 183,832.62 ± 13,647.79) (*t*-tests: Molino-Mante, $t_{791} = 3.12$, $p = 0.002$; Pachón-Rascón, $t_{791} = 2.31$, $p = 0.021$; Tinaja-Rascón, $t_{791} = 2.90$, $p = 0.039$). We found that the estimated age of the selective sweeps in these candidate genes (calculated with GEVA) did not differ across predicted mode of repeated evolution (from DMC) (Supplementary Fig. 15; Supplementary Data 5) and mostly dated to the approximate time of cave invasion (cave invasion estimated at 160–190 k generations before present; overlapping cave sweeps mean ± SD = 189,246 ± 28,255 generations before present; Supplementary Fig. 15).

We also used DMC to infer the mode of repeated evolution in a "control" set of 172 genes under selection in multiple surface populations but under neutral evolution in cavefish populations

(Supplementary Data 3). We observed maximum support (i.e., highest composite likelihood scores) for a model of repeated evolution via standing genetic variation in most surface genes (163 out of 172; 95%). Less than 5% of the surface genes analyzed were assigned the highest support by models of repeated evolution via independent mutations (3 out of 172; 2%) or migration (6 out of 172; 3%) (Fig. 4a). Thus, we observe that 20x more repeatedly adaptive sites evolved through de novo mutation in caves than between surface populations. The timing of when derived variants arose within selective sweeps (estimated with GEVA) in surface populations also differed from cave populations, with derived variants in surface sweeps being younger on average compared to cave sweeps and also showing higher variance around the mean (Lineage 1, Mante: mean ± SD = 130,754 ± 59,598 generations before present; Lineage 2, Rascón: mean ± SD = 127,077 ± 70,135 generations before present; Supplementary Data 4).

## Genes evolving repeatedly across cave lineages have a greater mutational opportunity

When the same genes are targeted in independent, repeated evolutionary events, it suggests that characteristics of those genes may predispose them to being drivers of phenotypic evolution. One leading hypothesis for molecular convergence is that longer genes experience more mutational opportunity[17]. We asked whether genes that have experienced repeated evolution across cave populations are longer than expected relative to the rest of the genomes. Indeed, our analysis revealed that the set of 760 overlapping sweeps genes had, on average, significantly longer coding sequence length (CDS with introns excluded), more exons, and more predicted transcript isoforms compared

to the reference database of all genes in the *A. mexicanus* genome (Fig. 4b). This same pattern was also observed for the set candidate genes identified by AF-vapeR as evolving repeatedly via locus reuse across seven cave populations (two-sided *t*-test, $t_{4373} = 18.843$, $P < 0.0001$; Supplementary Fig. 16). These patterns were not observed in the "control" set of 172 genes under selection in multiple surface populations but not in cavefish populations (Fig. 4c). Moreover, when broken down by the predicted mode of repeated evolution from our analysis with DMC, genes evolving repeatedly via independent mutations and standing genetic variation both had significantly longer transcript lengths compared to all genes in the annotated genome (Fig. 4d, f). Conversely, genes evolving repeatedly via migration among cave populations did not have longer transcript lengths compared to all genes in the annotation (Fig. 4e).

### Genes evolving repeatedly experience selection in potential regulatory regions

Lastly, we investigated whether repeated evolution is more likely to proceed via selection on inter- or intragenic regions. Our results suggest that the majority of repeatedly evolved changes under selection are likely regulatory in nature. Our DMC results indicated that the site under selection across all three cave populations was intragenic (between the start and stop coordinates) for 618 out of the 793 over-lapping sweeps candidate genes (Supplementary Data 5), and for 11 of these genes with site under selection was predicted to occur within the CDS. Five were present within the 5′ UTR and 15 were within the 3′ UTR. The remaining 587 intragenic sites were intronic. For the 175 genes with intergenic sites predicted to be the target of selection, the site was predicted to be within 10 kbp upstream in 97 genes and within 10 kbp downstream in 78 genes (Supplementary Data 5). This suggests that changes in gene regulatory elements that control gene expression (e.g., binding sites for regulatory proteins) could potentially be the target of selection nearly 80% of candidate genes for repeated evolution across caves.

We identified putative transcription factor binding site losses in cave populations relative to surface populations in 20 out of 175 genes where the site of repeated selection across caves was predicted to be intergenic (Supplementary Data 5). Several of the genes with a putative repeated transcription factor binding site loss have anno-tated phenotypes from Ensembl (v107) associated with regressive traits (e.g., *guca1b*, *uap1*, and *stoml1* are associated with abnormal optic disk morphology; Supplementary Data 5). Recently published data also indicate significantly higher expression of 12 of these gene in surface fish (Río Choy) compared to cavefish (Tinaja) (Supple-mentary Data 5), suggesting that the identified loss of a transcription factor binding site may have a functional impact. Future work is needed to functionally validate the repeated evolution of loss of transcription factor binding sites in these candidate genes in both cavefish lineages.

## Discussion

Repeated evolutionary events provide a powerful means for addres-sing some of the largest outstanding questions in evolutionary biology. Mutational screens in the laboratory show that there are many genetic paths to generate a functionally similar phenotype, yet, past studies have often observed the same mutations or reuse of the same genes underlying replicated trait evolution in natural populations[1,14,47] This suggests a higher degree of constraint in nature and/or that genetic changes observed in the wild may be particularly effective drivers of phenotypic change[14,17,18,48]. Here we leveraged the emerging model system of repeated evolution, *Astyanax mexicanus*, to address some fundamental questions about adaptative evolution. We showed that different alleles within the same gene (and less frequently the same allele within the same gene) are used repeatedly in the evolution of cavefish lineages. We also document that replicate evolved changes

between lineages largely rely on standing genetic variation and de novo mutations. Notably, we found that longer genes are more likely than others to be repeated targets of selection and that selection seems to have impacted fewer coding mutations than potentially regulatory mutations. Finally, we demonstrated genes associated with phenotypes that are commonly lost in caves are under selection. Each of these points are valuable to the *Astyanax* system, but also provide clear empirical examples for broader conclusions about repeated evolution and the potential adaptive nature of evolutionary loss[3,5,10,11,31,49]. Moreover, since the two cave forms originated from two separate lineages of surface fish, this system offers rare insight into repeated evolution experienced within a single species, rather than the sorting of recent ancestral alleles, which is most often seen within species[3].

While the Mexican cavefish system is used as model of repeated evolution for studying a wide variety of traits including metabolism, sleep and circadian rhythms, the number of independent origins of these phenotypes—or if there even were multiple origins of the cave phenotypes—is still actively under debate[35,50–52]. Clarifying the origins of the cave phenotypes has critical implications for the most basic interpretations of the work done in this emerging model system. Our more extensively sampled phylogenetic analyses demonstrates strong support for two lineages of surface *A. mexicanus*, which each gave rise to cave populations that evolved cavefish phenotypes. This is also supported by our allele reuse analysis (detailed below) that clearly demonstrates repeated evolution in this system, consistent with mul-tiple origins of the cave-derived phenotype. This work adds to a sub-stantial amount of previous literature demonstrating by a variety of marker types (e.g., allozymes, microsatellites, RADseq, mitochondrial data) that there are multiple, independent origins of the cavefish phenotype across the *Astyanax*-containing caves in Mexico[33–35,50,53–57] and provides a critical framework for future and past studies in this model system.

We refer to these two separate evolutionary branches as Lineage 1 and Lineage 2. Lineage 1 corresponds to the Gómez Farías and Chamal-Ocampo regions (Guatemala cave region) known in previous publica-tions as the "new" lineage. Lineage 2 corresponds to the El Abra cave region, previously known as the "old" lineage. Presently, most surface locations appear to be dominated by Lineage 1, except Gallinas and Rascón (which are within the same drainage) and Los Peroles. These three surface populations have been previously discussed to be his-torically isolated from other surface populations[35,58]. We prefer the "Lineage 1" and "Lineage 2" designations since Herman et al.[35] places the divergence of the caves of both lineages from their surface coun-terparts at approximately the same age (161–190 k generations ago), and thus, neither is a "new" or "old" cave lineage. Notably, our analysis of derived variant age within selective sweeps also supports this timing of the origin of cave-derived phenotypes. The method employed here accounts for gene flow and demography (i.e., GEVA) and places derived variants within selective sweeps for genes related to cave-derived traits near 150–190k generations ago for both lineages of cavefish (Fig. 2, Supplementary Fig. 10, Supplementary Data 4). Thus, our data do not support a more recent divergence between cave and surface fish[51].

Finally, while it appears that Subterráneo may be a third, inde-pendent origin of cave-derived phenotypes, we show that the cave ancestry present in this cave is most closely related to other Lineage 2 caves. This cave appears to have received so much gene flow from ongoing floods with Lineage 1 surface populations that it phylogen-etically clusters with the Lineage 1 surface populations, as first pro-posed by[56]. Thus, this cave is likely not an independent origin of cave-derived phenotypes, as would be suggested from examining the phy-logenetic tree alone (Fig. 1c). By coupling ancestry analysis with phy-logenetics using whole genome sequence data, we were able to generate a more informed hypothesis of the origins of this cave

population, and an overall understanding of independent origins of the *A. mexicanus* cave phenotype.

In *On the Origin of Species*, Darwin postulated that loss of eyes in cave animals was likely due to disuse and drift rather than direct selection against eyes[26]. The idea that regressive trait evolution is largely driven by disuse and drift remains prevalent in the literature today[24], and is also a matter of great contention within the cavefish literature[25]. However, the data presented here suggests that signatures of selection present across cavefish, but not in surface fish populations, are found in genes associated with regressive traits (e.g., loss of pigmentation, sleep, and eyes) (Fig. 2, Supplementary Data 2, 3, 7). The hypothesis that disuse and relaxed selection are driving the regressive cave-derived phenotypes is not supported by several other observations, as well. First, recurrent loss of traits is observed even across very distantly related cave taxa[23,59]. This phenomenon is unlikely to have occurred repeatedly through drift alone. Second, eyesight, image processing, and circadian rhythms are energetically expensive[60,61], so it is plausible that loss of eyes in a dark environment could confer a fitness advantage and be selected for directly. Third, previous work in this system has provided evidence of selection via pleiotropy, with indirect selection on regressive traits resulting from direct selection on constructive traits[44,62,63], and our results indicate that many genes under selection in cavefish populations are likely pleiotropic (Supplementary Fig. 10, Supplementary Data 2, Dryad repository, https://doi.org/10.5061/dryad.3xsj3txmf). Fourth, we previously implemented the 12-locus additive alleles model by Cartwright et al.[64] with parameters estimated for Molino cave (as this is one of the populations where selection would need to be strongest to overcome the effects of drift due to small population size). This analysis showed that within the time period since invasion of the caves, with secondary contact and gene flow between cave and surface populations, positive selection of moderate strength is needed to drive alleles that produce a blind fish to fixation[35]. Indeed, the population-specific and repeated sweeps across cave populations presented here clearly demonstrate that positive selection played a substantial role in shaping cave-derived constructive and regressive phenotypes.

We note that while our results suggest a role for selection in regressive trait evolution, we absolutely expect that neutral evolution (and specifically genetic drift) also contributes to regressive evolution, and a recent study found little evidence of positive selection in *Astyanax* cavefish using divergence-based approaches limited to coding regions in one cave population (Pachón)[65]. This contrasts our finding that more of the genome had evidence of selective sweeps in cavefish compared to surface fish populations (Supplementary Fig. 7, Supplementary Table 2). Population bottlenecks can cause a pattern of reduced diversity that can be erroneously interpreted as a sweep, but the approach we used to detect selection considered demographic events and is therefore robust against such errors[37]. We also present evidence here suggesting that non-coding regions are frequently the target of selection (Supplementary Data 5), which would not be detected by divergence-based approaches that rely on quantifying substitution rates in coding regions. While speculative, the within-population level tests that our work implements may be more sensitive than between species tests implemented by Zhou et al.[65], and these different approaches may be identifying signatures of selection corresponding to a different stage of the evolutionary process.

Whether we can expect adaptive evolution to follow similar or variable paths when replicate populations are exposed to similar selective pressures is a fundamental question in biology. Our scans for repeated selection in two lineages of cavefish populations indicate that reuse of the same locus (13% of all genes) is much more common than reuse of the exact same allele (0.13% of all genes) (Supplementary Data 6). This suggests that adaptive evolution has largely followed unique molecular trajectories to achieve similar phenotypic endpoints across lineages in this system and agrees with our finding that repeated

evolution is rarely attributed to gene flow between lineages in *A. mexicanus* (Supplementary Data 5). A recent examination of the genetic basis of repeated evolution at a deeper evolutionary timescale found no evidence of overlap in positively selected genes among three different distantly related cavefish species[65]. This contrasts our finding that the same loci are often targeted repeatedly by selection in both *A. mexicanus* lineages which provides support to the hypothesis that historical contingency plays a more important role in repeated evolutionary events between more closely related taxa[9]. We speculate that the differences in conclusions may be attributable to between-species comparisons examining deeper time scales than the within-species comparisons employed in our work.

Notably, we find that selection on standing genetic variation accounts for about 53% of shared sweeps among three focal cave populations, while de novo mutations contributed to 40% of shared sweeps, and migration accounted for only 7% of shared sweeps. A general pattern that has emerged from genomic surveys of repeated evolution is that selection on standing genetic variation and alleles transferred by gene flow are more likely to underlie repeated trait evolution within species, whereas independent mutations are more likely to underlie repeated trait evolution between more distantly related taxa (reviewed in ref. 3). Our findings are among the most thorough empirical demonstration of how both processes may be important within single species.

Our finding that 40% of the candidate genes for repeated evolution across caves have evolved via de novo mutations is surprising given that these populations adapted to caves relatively recently. Rapid adaptive evolution is often attributed to allele reuse through selection on standing genetic variation or gene flow[66] (e.g., songbirds[67,68], fishes[68-73], insects[74], and plants[9]). Conversely, adaptive evolution through de novo mutations is typically thought of as a slower process, making it notable that unique mutations appear to have occurred within the same genes multiple times independently in the recent past in both cavefish lineages. In cases where rapid adaptation has been attributed to de novo mutations, loss-of-function mutations are often implicated[2,75,76]. A unique aspect of the cavefish system is that loss-of-function mutations may be particularly likely to be adaptive, allowing for the cavefish system to be a unique situation where a recent adaptation can be impacted substantially by de novo mutations. Additionally, founding populations in caves may have been large (i.e., in the case of stream capture[77,78]) and therefore may not have been mutation-limited[79].

While future work remains to empirically quantify direct mutation rates between cave and surface fish, the prevalence of de novo mutations driving adaptive evolution in this system may suggest elevated mutation rates in cavefish, possibly due to some unique qualities of cavefish genomes and physiology and/or the cave environment. Multiple lines of evidence support a hypothesis that mutation rates may be higher in cavefish populations compared to surface fish populations. First, mutation rates can increase in novel, stressful environments, such as caves[79-81]. Second, recombination rates can also increase in new, stressful environments, and recombination can be mutagenic[82,83]. Third, the cavefish genome experienced a recent increase in transposable elements (Tc-Mariner and hAT superfamilies) around time of cave entry[84], and this wide-spread expansion may have resulted in adaptive mutations much like in the peppered moth adapting to industrial pollution[85]. Fourth, DNA repair genes are upregulated in cavefish[86,87]. Though it is currently unclear what impact this may have on mutation rate and whether this upregulation occurred in response to elevated DNA damage in hypoxic cave environments, DNA repair can introduce mutations[88], so this is also suggestive of shifts in mutation rate or profile among cavefish relative to surface fish. Finally, the cave habitat itself might predispose cavefish to higher mutation rates relative to surface populations. For example, the radioactive gas, radon, often accumulates in caves and ground water and has been

linked to elevated mutation rates in cave-dwelling crickets[89]. Thus, we hypothesize that cavefish populations may experience elevated mutation rates compared to surface populations and suggest that these factors may be sources of mutational input in this system.

Finally, repeated evolution of mutations in the same genes in different lineages suggests these genes may have specific characteristics that lend them to be drivers of phenotypic evolution. Some hypotheses include that loci used often in phenotypic evolution may exhibit less deleterious pleiotropic consequences, are located in a certain part of a pathway[16,18], are within regions of the genome predisposed to faster evolution or more mutations (e.g., high recombination regions, fragile sites), or that longer genes may experience more mutations. While the first scenarios are difficult to comprehensively test, our data do support the notion that longer genes may contribute more to repeated evolution (Fig. 4b). Candidate genes for repeated evolution across cave lineages are longer compared to the genomic background, supporting the target size hypothesis for why certain genes contribute more often to repeated evolution than others. In contrast, genes with evidence of sweeps in both surface lineages do not differ in length from the rest of the genome (Fig. 4c). Further, genes evolving repeatedly in cave populations due to selection on de novo mutations are in longer genes compared to those evolving repeatedly via migration. While sweeps from standing genetic variation are in genes that are longer than the rest of the genome as well, this effect size is 2-fold less compared to genes with de novo mutations (Supplementary Table 8).

In conclusion, we leveraged whole genome sequencing to address fundamental questions regarding the evolution of repeated adaptation to cave environments in *A. mexicanus* and conclude that (1) two well-supported origins of the cave phenotype are independently derived from two distinct lineages of surface fish ancestors, and a previously proposed third potential origin is unlikely (and was the result of a misleading phylogenetic signal); (2) there is strong evidence that regressive cavefish phenotypes could not have evolved purely through disuse and drift; (3) strong selection in the cave environment caused adaptive traits to evolve nearly simultaneously upon fish entering caves and evolution of cave-derived traits was highly polygenic; (4) repeated evolution of the same traits between cavefish lineages proceeded predominantly via selection on standing genetic variation and de novo mutations, and we find little support for repeated evolution via migration between Lineage 1 and Lineage 2 cavefish populations; and (5) genes recurrently under selection across cavefish lineages are longer compared to the rest of the genome, a pattern primarily driven by genes evolving repeatedly via de novo mutation. These answers are crucial to inform future studies of development, plasticity, behavior, and neurobiology in this system and inform our understanding of how evolution proceeds.

## Methods

### Sample collection and sequencing

The ethical treatment of animals collected for this study was in compliance with Secretariat of Environment and Natural Resources permits SGPA/DGVS/2438/15, SGPA/DGVS/2438/16, SGPA/DGVS/05389/17, SGPA/DGVS/05389/18, and SGPA/DGVS/1893/19 to P. Ornelas-García. We obtained whole genome sequence data from a total of 248 *A. mexicanus* individuals sampled across eight surface and 18 caves populations, as well as four outgroup individuals (two *A. aeneus* and two *A. nicaraguensis*) (Fig. 1b; Supplementary Data 1). We sequenced 184 of these samples for the present study (Supplementary Data 1). We also obtained previously sequenced samples from ref. 35 (*n* = 48) and[90] (*n* = 21) (Supplementary Data 1).

### Genotyping

For Caballo Moro samples, we used Trimmomatic v0.30[91] to remove adapters and perform quality trimming. For all other samples, adapters were trimmed using Cutadapt v1.2.1[92] with barcodes specified for each individual when available. We used the wildcard option in Cutadapt for samples with unknown barcodes (i.e., samples sequenced at BGI; see Supplementary Data 1). Samples were then trimmed for quality using Trimmomatic v0.30. We required a minimum quality score of 30 across a 6 bp sliding window and discarded reads with a length of <40 nucleotides.

Reads were aligned to the surface *Astyanax mexicanus* genome (Astyanax_mexicanus-2.0 GCF_000372685.2 [https://www.ncbi.nlm.nih.gov/assembly/GCF_000372685.2/])[93] using bwa v0.7.4. We used Picard v2.3.0 (http://broadinstitute.github.io/picard/) to remove duplicates and add read group information and used samtools v1.7[94] to split de-duplicated bams into mapped and unmapped reads.

We conducted genotype calling following the GATK Best Practices[95] (Supplementary Table 9). Mapped bams were used to generate per-individual gvcfs with the Genome Analysis Tool Kit (GATK) v3.7.0 HaplotypeCaller tool. We used the GenotypeGVCFs tool in GATK v3.8.0 to produce vcf files for each chromosome and unplaced scaffolds that include all individuals (and include invariant sites). The SelectVariants and VariantFiltration tools in GATK v3.8.0 were used to apply hard filters. We subset vcfs for each chromosome and unplaced scaffold into invariant, SNP, and mixed/indel sites and applied filters separately following GATK best practices (Supplementary Table 1). We then used the MergeVcfs tool in GATK v4.1.4 to recombine all subset VCFs for each chromosome and unplaced scaffold. All scripts used in QC and genotyping are available at https://github.com/rachelmoran28/cavefish_2019_pipeline. We conducted phasing with Beagle v5.1[96]. Indels and the 3 bp region around each indel were removed using a custom python script (available at https://github.com/rachelmoran28/cavefish_2019_pipeline). We used the vcftools (v0.1.15)–exclude-bed option to remove repetitive regions identified by WindowMasker and RepeatMasker with files downloaded from NCBI (*Astyanax mexicanus* Annotation Release 102 GCF_000372685.2 [https://ftp.ncbi.nlm.nih.gov/genomes/all/GCF/000/372/685/GCF_000372685.2_Astyanax_mexicanus-2.0/]). We also used vcftools to only retain biallelic SNPs, to remove sites with greater than 20% missing data within each population, and to remove variants with a minor allele frequency <1%. We calculated heterozygosity at each site for each population in R (v3.6.3) using the vcfR package. We removed sites where every sample in a given population was heterozygous, indicative of collapsed paralogs. This resulted in retaining a total of 287,966,961 sites throughout the genome, 14,742,459 (5.12%) of which were biallelic SNPs (Supplementary Data 8).

### Population genomic summary statistics

We calculated absolute genetic divergence (Dxy) and relative genetic divergence (Fst) between each pair of populations and nucleotide diversity (Pi) within each population (Supplementary Figs. 17, 18) in non-overlapping 50 kb windows across the genome using the python script popgenWindows.py (https://github.com/simonhmartin/genomics_general/blob/master/popgenWindows.py).

We also calculated Dxy and Fst between populations and Pi within populations on a site-by-site basis using a custom python script (available at https://github.com/rachelmoran28/popgen_stats_by_gene) (Supplementary Data 9). This allowed us to calculate summary statistics (e.g., minimum, maximum, and mean values) for each of these metrics across the coding and untranslated region of each gene in the surface fish *A. mexicanus* 2.0 genome annotation (Ensembl v101, downloaded from ftp://ftp.ensembl.org/pub/release-101/gtf/astyanax_mexicanus/).

### Population structure

To examine population structure in our data set, we conducted Principal Component Analysis (PCA) and ADMIXTURE (v1.3.0) analysis. To better detect fine-scale patterns of genetic variance among closely

related lineages within *A. mexicanus*, we excluded the outgroup individuals for the PCA. We used Plink v1.90 to prune SNPs in 50 kb windows with a 10 bp window step size and specified a linkage threshold of $r^2 < 0.1$. We ran a PCA in Plink v1.90 on the resulting pruned set of 751,759 SNPs.

For the ADMIXTURE analysis, we included the two *A. aeneus* outgroup samples and filtered the original set of SNPs to remove sites with >10% missing data and thinned to 1 SNP per 50 kb. To further ensure that SNPs used in the ADMIXTURE analyses did not co-occur within close proximity on the same chromosome, we did not use SNPs found on unplaced scaffolds. This filtering resulted in a set of 9717 SNPs. Binary file sets were generated with Plink v1.90. Cross-validation was run for K 1–15.

## Phylogeography

Understanding the phylogenetic relationship between study populations is critical to interpreting population genomic analyses and tests for selection. A phylogenetic approach can also be used to identify signatures of repeated evolution. For example, identifying the number of instances where surface and cavefish lineages are inferred to be sister taxa in a phylogenetic tree can provide insight into how many times cave-adapted phenotypes have evolved from ancestral surface lineages. We used Maximum Likelihood (ML) and coalescent approaches with SNPs and gene trees to infer the phylogenetic relationship among *A. mexicanus* populations and investigate phylogenetic patterns consistent with the repeated evolution of cave adaptation. The two *A. aeneus* samples were included as an outgroup for all phylogenetic analyses.

First, for the SNP-based analyses, the dataset was thinned to 1 SNP per kb, resulting in 680,021 SNPs throughout the genome. We estimated a ML population tree in Treemix v1.13. We used the script vcf2treemix.py (https://github.com/CoBiG2/RAD_Tools; accessed 3/6/2020) to convert the vcf containing thinned SNPs to a tmix file for the Treemix analysis. We estimated a species tree using the multi-species-coalescent in SVDquartets, implemented in PAUP v4.0a. To prepare our data for SVDquartets, we converted the vcf of thinned SNPs to a nexus file using the ruby script convert_vcf_to_nexus.rb (https://github.com/mmatschiner/tutorials/blob/master/species_tree_inference_with_snp_data/src/convert_vcf_to_nexus.rb; accessed 4/2/2022). We ran SVDquartets specifying a sampling of 500,000 random quartets and 500 standard bootstrap replicates. Both Treemix and SVDquartets assume SNPs are unlinked and have independent evolutionary histories.

Second, we inferred a coalescent-based gene tree in ASTRAL-MP (v5.15.4)[97]. We identified 3,442 single copy orthologs in the *Astyanax mexicanus* surface fish genome (v2.0) using the Benchmarking Universal Single-Copy Ortholog assessment tool (BUSCO) v2.0[98]. We supplied the Actinopterygii Odb10 lineage and zebrafish for gene predictor training with AUGUSTUS (v3.3.3)[99]. We used the python script vcf2msa.py (https://github.com/tkchafin/vcf2msa.py; accessed 4/2/2022) to generate fasta alignments for each of the 3442 single copy orthologs from a vcf containing invariant sites and SNPs for all 246 samples. ML trees were built for each gene using IQ-TREE (v1.6.12)[100]. These trees were concatenated and supplied to ASTRAL-MP for species tree construction.

## Determining the origins of Cueva del Río Subterráneo

Recent introgression events can have a large impact on species tree inference. We recently found evidence that Chica and Caballo Moro caves contain surface-cave hybrids with 15–25% of their genomes derived from surface fish ancestry, on average[90], Medley et al. unpublished data). Previous studies have suggested that Subterráneo cavefish may also experience a substantial amount of admixture with the local surface population. Preliminary phylogenomic analyses indicated that Subterráneo grouped with surface populations rather

than with other cave populations, and therefore may represent a unique evolutionary origin of cave adaptation distinct from the El Abra and Guatemala cave lineages. However, this pattern might also be driven by substantial ongoing gene flow with the local Micos River surface population.

Surface water floods into Subterráneo cave during the wet season, bringing *A. mexicanus* surface fish into the cave. Unlike most other cave populations (with the exception of Chica and Caballo Moro hybrid populations), cavefish within Subterráneo have been shown to have eyes present (although reduced in size compared to surface fish) and pigmentation[78,101]. Surface fish have also been documented in Subterráneo cave during the wet season after high water. Although previous phylogenetic analyses have found support for Subterráneo cave as a close relative to the Lineage 1 cavefish in the Guatemala region, this cave occurs within El Abra limestone and is geographically closer to the El Abra caves compared to the Guatemala caves. Ongoing hybridization with Lineage 1 surface fish (i.e., from Arroyo La Pagua that floods into the cave,[78]) could be masking that this cave population originated from the same Lineage 2 surface fish stock that populated the El Abra caves (first proposed by ref. 56).

To explore this hypothesis, we conducted formal tests for introgression between Subterráneo cavefish, a Lineage 1/Guatemala region cavefish population (Escondido), two Lineage 2/El Abra region cavefish population (Pachón and Tinaja, representing caves at the northern and southern extent of the El Abra cave region), a Lineage 1 surface fish population (Mante), and a Lineage 2 surface fish population (Rascón). Mante was included as the Lineage 1 surface fish population in these analyses rather than Micos due to sample size (Micos: $n = 1$; Mante: $n = 10$). Two *Astyanax nicaraguensis* samples served as an outgroup comparison.

We conducted formal tests for introgression with Treemix and using D and f4 statistics. We first used Treemix v1.13[102] to visualize migration events and confirm phylogenetic relationships between the Subterráneo population and the two non-admixed cave populations, the two surface populations, and the outgroup. Treemix builds a bifurcating tree to represent population splits and also incorporates migration events, which are represented as "edges," connecting population branches. For this analysis, we used biallelic SNPs thinned to 1 kb apart. We supplied the resulting set of 1,148,321 SNPs to Treemix, rooted with *A. nicguensis*, and estimated the covariance matrix between populations using blocks of 500 SNPs. Sample Rascón _6 was excluded from this analysis because ADMIXTURE indicated that it was likely an early generation hybrid. We first built the maximum likelihood tree (zero migration events) and then ran Treemix sequentially with one through six migration events. We calculated the variance explained by each model (zero through six migration events) using the R script treemixVarianceExplained.R[103].

We used Dsuite v0.4[104] to conduct formal tests for introgression between Subterráneo cavefish and Lineage 1 and Lineage 2 surface and cave populations. This allowed us to further test the hypothesis that Subterráneo represents a hybrid population resulting from admixture between a Lineage 2/El Abra region cave population and a Lineage 1 surface population. If gene flow has occurred between Subterráneo cavefish and the local Lineage 1 surface population, we predict an excess of shared derived alleles between Mante and Subterráneo. This analysis may also provide insight into which lineage of surface stock founded the Subterráneo cave population. If Subterráneo cave was initially populated by Lineage 2 surface fish, we would expect Subterráneo cavefish to share more derived alleles with Lineage 2 cave populations compared to Lineage 1 cave populations. However, we note that recent, ongoing introgression with the Lineage 1 surface fish may artificially inflate the number of shared derived alleles between Subterráneo and Lineage 1 cavefish.

We supplied the same set of 1,148,321 thinned biallelic SNPs used in the Treemix analysis to Dsuite and specified *A. nicaraguensis* as the

outgroup. We again excluded the one sample from Rascón with apparent hybrid ancestry. We used the Dsuite program Dtrios to calculate Patterson's D statistic for all possible trios of populations using the ABBA-BABA test[104]. The ABBA-BABA test quantifies whether allele frequencies follow those expected between three lineages (e.g., sister species P1 and P2, and a third closely related species, P3) under expectations for incomplete lineage sorting (ILS). Observing a greater proportion of shared derived alleles between P1 and P3 but not P2 or between P2 and P3 but not P1 than what would be expected by chance (i.e., ILS) indicates introgression. Dsuite requires a fourth population, P4, to serve as an outgroup and determine which alleles are ancestral versus derived. Ancestral alleles are labeled as "A" and derived alleles are labeled as "B". ABBA sites are those where P2 and P3 share a derived allele, and ABAB sites are those where P2 and P4 share a derived allele. The D statistic is calculated as the difference in the number of ABBA and BABA sites relative to the total number of sites examined. Dsuite uses jackknifing of the null hypothesis that no introgression has occurred (D statistic = 0) to calculate a $p$-value for each possible trio of populations.

Dsuite also calculates the admixture fraction, or f4-ratio, which represents the covariance of allele frequency differences between P1 and P2 and between P3 and P4. If no introgression has occurred since P1 and P2 split from P3 and P4, then f4 = 0. If the f4 statistic is positive, this suggests a discordant tree topology indicative of introgression.

We then quantified introgression across the genome in Subterráneo cavefish using Hidden Markov Model (HMM) and fine-scale SNP mapping approaches to calculate ancestry proportions globally (i.e., genome-wide averages) and locally (i.e., at each site along each of the 25 chromosomes). We implemented a HMM-based approach in Loter to infer genome-wide local ancestry in the Subterráneo individuals. Mante served as the parental surface population for the initial training stage of the HMM, as we only had a sample size of one for the local Micos surface population and our phylogenetic analyses revealed that Mante surface fish are closely related to the Micos and Subterraneo populations. As preliminary analyses indicated that Subterráneo shared more derived alleles with Lineage 2/El Abra region cavefish compared to Lineage 1/Guatemala region cavefish, we ran the analysis with Pachón as the proxy for the parental cave population (i.e., representing the genome of Subterráneo cavefish prior to onset of recent admixture with the local surface population). This analysis allowed us to estimate global ancestry proportions and mean minor and major parent tract lengths for each individual. Ancestry tract lengths were converted from base pairs to Morgans using the median genome-wide recombination rate 1.16 cM/Mb (0.0000000116 Morgan/bp) obtained from a previously published genetic map for *A. mexicanus*[63]. We then estimated the number of generations since the onset of admixture ($T_{admix}$) using Eq. (1):

$$T_{admix} = 1/(L_M * p_B) \tag{1}$$

where $L_M$ is the mean ancestry tract length from the minor parent in Morgans and $p_B$ is the proportion of the genome derived from the major parent (the probability of recombining)[64–66]. This analysis indicated an estimated mean ± SE of 9843 ± 600 generations since the onset of admixture.

### Selection analyses

We used a convoluted neural network approach implemented in diploS/HIC to identify regions of the genome that show evidence of selective sweeps. diploS/HIC generates multiple population genetic statistics to infer selection and classifies genomic windows as being neutral, hard sweeps, soft sweeps, or linked to a region that experienced a hard or soft sweep. The convoluted neural network is first trained and tested using simulated population genetic data. The resulting model is then applied to make predictions on empirical data.

This machine learning approach has been shown to be highly robust to demographic model misspecification[37]. However, due to substantial differences in historical and present-day population sizes between cave and surface populations, we chose to generate two separate sets of simulated population genetic data, one for cave populations and one for surface populations, for use in training diploS/HIC's convoluted neural network. Results were qualitatively similar when the same training data was used to predict regions under selection across both surface and cave populations, demonstrating that diploS/HIC's predictions are not heavily influenced by imperfect demographic model specification.

We simulated population genetic data with neutral, soft, and hard sweeps in discoal[36] using cave- or surface-specific demographic parameters estimated from Stairway Plot 2 (see Supplementary Note 2, Fig. 2a; surface model: present-day population size = 5,263,992 with historical population size changes -en 1.250000 0 0.045000 -en 1.875000 0 0.250000; cave model: present-day population size = 189,942 with historical population size changes -en 0.999685 0 0.037788 -en 4.152884 0 2.098659). We ran 3000 simulations with 40 chromosomes, as this diploid chromosome number is close to our population sample sizes for our data and specified a sequence length of 55 kb (to match the window size used with the actual data in diploS/HIC; see below). We specified a mutation rate of 3.5e-9, obtained from data on cichlids[104]. For hard and soft sweep simulations, we specified priors on tau (time since fixation, in units of $4N_0$ generations ago) ranging from 0 to 0.05. For soft sweep simulations, we specified priors on $f_0$ (initial selected frequency) ranging from 0 to 0.10. These were the default parameters and seemed to give the best results upon exploration of runs with different parameters. We ran simulations of selective sweeps where the sweep occurred in each of 11 equidistant locations along the 55 kb sequence.

The simulated surface and cave datasets were used to generate two sets of feature vectors in diploS/HIC for training and testing the convoluted neural network. Empirical data was then provided to diploS/HIC to generate feature vectors and predictions. As with our simulations, we generated diploS/HIC predictions on our empirical data in 11 subwindows across larger 55 kb windows for each chromosome or unplaced scaffold. Thus, diploS/HIC generated predictions on 5 kb windows across the genome, classifying each window as neutral, soft, hard, softLinked, or hardLinked. Some windows were skipped by diploS/HIC due to missing data or lack of SNPs (see Dryad repository, https://doi.org/10.5061/dryad.3xsj3txmf).

We conducted additional genome scans for selection using hapFLK[105]. This Fst-based test takes into account hierarchical population structure and haplotype structure[105,106]. However, hapFLK has less sensitivity to detect soft sweeps (i.e., selection on standing genetic variation) compared to diploS/HIC. hapFLK is among the most powerful methods when tested to detect positive controls[106], though it may be sensitive to extreme bottlenecks and migration. We included a single *Astyanax aeneus* to serve as an outgroup, and we analyzed the Lineage 1 and Lineage 2 populations in separate hapFLK runs.

We identified nonsynonymous coding variants across all populations and predicted the consequence of each variant on protein function using computational analysis with the SIFT (sorting intolerant from tolerant) algorithm[107] and the Ensembl Variant Effect Predictor (VEP) software suite using Ensembl v100 annotations[108]. SIFT uses sequence homology and data on the physical properties of a given protein to predict whether an amino acid substitution will be tolerated or deleterious. VEP performs annotation and analysis of genomic variants to predict impact on the protein sequence (i.e., modifier, low, moderate, or high) (Dryad repository, https://doi.org/10.5061/dryad.3xsj3txmf).

### Inferring adaptive alleles within cave populations

We next wanted to investigate the traits that were most important in initially facilitating cave adaptation. To this end, we asked whether

genes with annotations associated with different cave-derived phenotypes experienced selective sweeps at different times (e.g., whether metabolic genes experienced selective sweeps prior to pigmentation genes). Within a given cave population, adaptive alleles were defined as genes with evidence of a selective sweep in the cave population (identified with diploS/HIC) and no evidence of a sweep in a same-lineage surface population (neutral or linked calls from diploS/HIC). Although diploS/HIC offers a powerful approach to detect regions of the genome containing sweeps versus those that do not, it may not be as reliable at distinguishing hard sweeps from soft sweeps[37]. For this reason, we did not differentiate between hard and soft sweeps for this analysis (i.e., putatively adaptive alleles could contain a hard or soft sweep in a given cave population).

We used this approach to identify adaptive alleles in seven total cave populations. This included three Lineage 1 populations (Molino, Vasquez, and Caballo Moro), and four Lineage 2 populations (Pachón, Tinaja, Yerbaniz, and Palma Seca). We only included cave individuals that were eyeless and had no evidence of recent admixture with surface populations in our preliminary analyses. We used Mante as the surface population in Lineage 1 comparisons and Rascón as the surface population in Lineage 2 comparisons. The focal cave and surface populations were chosen due to their relatively high sample size and coverage relative to the entire dataset (Supplementary Data 1).

To test for differences in the onset of selection between functional categories of adaptive traits, we focus our analysis of selective sweep ages on a subset of genes in each population with associated GO terms containing keywords (e.g., pigment, sleep, neuromasts) corresponding to traits that are derived in cave habitats. We obtained GO terms associated with each gene in the *A. mexicanus* surface fish annotation from Ensembl's Biomart (v104 [https://www.ensembl.org/biomart/martview/]) (Dryad repository, https://doi.org/10.5061/dryad.3xsj3txmf). We then used a list of keywords associated with each phenotypic category to pull out genes from the list of candidate adaptive genes in each cave population (Supplementary Table 5). As defined above, genes that are candidates for adaptation were classified as having a soft or hard sweep in a given cave population but no sweep in a same-lineage surface population.

### Estimating the age of selective sweeps in adaptive alleles

We took two approaches to test for temporal differentiation in cave adaptation across functional categories. First, for each putatively adaptive allele, we estimated the time since the last common ancestor between cave and surface alleles T, using absolute sequence divergence between populations (Dxy) and the per-bp, per-generation mutation rate ($\mu$, estimated from cichlids as $3.5 \times 10^{-9}$;[104]) using Eq. (2).

$$Dxy = 2*\mu T \qquad (2)$$

This approach provided an absolute estimate of when the cave allele split from the surface allele.

Second, we used a recently developed nonparametric approach, Genealogical Estimate of Variant Age (GEVA)[40], to obtain relative estimates of when derived variants present in selective sweeps occurred in genes containing adaptive alleles. This approach incorporates demographic information in addition to mutation and recombination rate and uses coalescent modeling to infer the age of alleles that are putatively under selection relative to the most recent common ancestor. Population genomic data for an entire chromosome can be supplied to GEVA along with the location of the variant of interest. The ancestral segment of the allele on which the mutation has occurred is inferred using an HMM. We specified a mutation rate of $3.5 \times 10^{-9}$ estimated from cichlids[104], a recombination rate of $1.16 \times 10^{-6}$ cM/bp estimated from the median genome-wide recombination rate from a previous *A. mexicanus* linkage map[109], and a generation time of 1 year[35]. Initial exploration of our data using Shapiro-Wilk tests revealed that

data in all populations did not fit a normal distribution. We therefore used non-parametric Kruskal-Wallis rank sum tests to ask whether the mean estimated timing of derived variants within selective sweeps differed across phenotypic categories.

### Identifying candidate genes for repeated evolution between cave lineages

We identified regions of the genome that have undergone repeated evolution between lineages using two methods. First, we conducted scans for repeated selection across replicate surface-cave population pairs with AF-vapeR (Allele Frequency Vector Analysis of Parallel Evolutionary Responses)[7]. Rather than scanning each replicate cave population for local selection, this approach uses eigen decomposition over allele frequency change vectors (here in comparisons between same-lineage surface and cave populations) to detect and classify loci experiencing different categories of repeated selection (i.e., full parallel, antiparallel, multiparallel, or divergent). For this analysis, we included all seven cave populations used in the sweep dating analysis (three Lineage 1 and four Lineage 2) and two surface populations (one Lineage 1 and one Lineage 2; see above). While AF-vapeR does not explicitly test for selection and other evolutionary scenarios may lead to parallel allele frequency change[7], we are making the assumption that highly parallel allele frequency change is an indicator of selection, and classify loci as experiencing full parallel selection if they show allele frequency changes in the same direction at the same site across all populations (i.e., all cave populations are moving in the same direction away from surface populations). We refer to this as "allele reuse". Anti-parallel selection is classified as allele frequencies moving in different directions at the same locus (i.e., allele frequencies in some cave populations are diverging from surface populations and others are becoming more similar to surface populations). Under multiparallel selection, repeated evolution is occurring along multiple trajectories at the same locus (i.e., in both cave lineages the same locus is diverging from surface populations but unique alleles are selected for in each lineage). Thus, in the case of multiparallel selection, repeated evolution is occurring within each lineage independent of the other lineage (*sensu* convergent evolution). We refer to this as "locus reuse". Finally, divergent selection occurs when multiple populations are diverging from one another via selection on unique alleles at the same locus (i.e., multiple different alleles are present at a given locus within Lineage 1 and Lineage 2 cavefish, and allele frequencies in cave populations differ from those present in surface populations).

For the present study, we were interested in investigating genomic regions under full parallel and multiparallel selection (i.e., allele reuse and locus reuse, respectively). We first identified loci where all seven cave populations examined show divergence from surface populations via selection on the same allele (allele reuse), which are expected to have high loadings on eigenvector 1 and lower loadings on subsequent eigenvectors (visualized as a peak on eigenvector 1). Second, we identified loci where two cavefish lineages show evidence of selection for unique alleles (locus reuse), which are expected to have high loadings on eigenvector 1 and eigenvector 2 (visualized as a peak on eigenvector 2).

We used AF-vapeR to scan the genome for patterns of repeated allele frequency changes between surface and cave populations in windows of 50 SNPs (median physical window size 7.6 Kbp). We assessed significance using empirical *p*-vales compared to 10,000 null permutations. Significant windows were classified as having eigenvalues above the 99th percentile (empirical *p*-value < 0.01). We identified all significant windows with evidence of allele reuse across both lineages or locus reuse (unique trajectories of selection within each lineage). Genes within each significant allele reuse window (34 genes) and locus reuse window (3590 genes) were identified using the *A. mexicanus* surface fish genome annotation.

We then took a second independent approach to identify candidate loci for repeated evolution by detecting overlapping selective sweeps across cave populations. We chose to focus this analysis on the three cave populations with the highest sample sizes and coverage, Pachón, Tinaja, and Molino. These three populations also are among the most well-studied of the *A. mexicanus* cave populations with well-maintained laboratory stocks. Molino is a Lineage 1 cave and Pachón and Tinaja are both Lineage 2 caves. However, Pachón cave is ~60 km north-northwest of Tinaja cave, and recent population genomic analysis (including the present study) indicate that Pachón may be following a unique evolutionary trajectory and could be somewhat isolated from the other El Abra caves. Thus, including these three cave populations allowed us to compare patterns of repeated evolution between and within lineages.

We identified loci that were classified by diploS/HIC as having a soft or hard selective sweep in all three cave populations and no sweeps (i.e., neutral or linked calls) in surface populations. Genes were selected as putative candidates for repeated evolution if there was evidence of a soft or hard selective sweep between the start and stop coordinates from the gff, including introns, exons, and UTRs (760 genes). Because diploS/HIC classified regions in 5 kb windows across the genome, windows occurring near the end of the coding region of a gene could pick up on a sweep due to a selected variant in an upstream non-coding regulatory region, and the gene itself would still be classified as having a selective sweep. Although this approach may cause us to miss some candidates for repeated evolution of phenotypes among caves that are evolving via unique genes in different cave populations, identifying genomic regions under selection in multiple cave populations was required for our analysis on the mode of repeated evolution (see below).

We used permutation tests to ask whether regions of the genome containing candidate genes for repeated evolution among cave lineages overlap more than expected by chance with previously identified QTL regions associated with cave-derived phenotypes. We also asked whether candidate genes for repeated evolution of cave phenotypes are enriched for genes that are were previously shown to be differentially expressed in surface versus cavefish populations, relative to the rest of the genome.

Lastly, we investigated whether the set of genes we identified as putatively being involved in repeated evolution across cave lineages were enriched for certain features previously implicated in genetic changes underlying cave adaptation in this system and in instances of repeated evolution in general. Longer genes may be more likely to be the target of repeated de novo mutations across lineages by proving a larger mutational target. Thus, we tested whether candidate genes for repeated evolution had longer coding sequence length (CDS) and transcript length. We also used SIFT[107,110] and VEP[108] to identify putatively deleterious mutations in each population for each annotated gene (see Dryad repository, https://doi.org/10.5061/dryad.3xsj3txmf).

Furthermore, cavefish populations exhibit regressive traits that may be associated with large-effect mutations that could confer loss of function. Accordingly, we examined candidate genes for loss of transcription factor binding sites in cave populations relative to surface populations.

The complete set of vertebrate non-redundant transcription factor binding motifs were downloaded from the JASPAR CORE (9th release)[111], totaling 838 motifs. The FIMO tool (v5.0.1)[112] from the MEME Suite was used to scan for these motifs 10 kb upstream and downstream of candidate genes for repeated evolution. We focused on 760 genes identified with overlapping selective sweeps in both cave lineages (as described above). We scanned two surface populations (Lineage 1: Río Choy; Lineage 2: Rascón;) and three cave populations (Lineage 1: Molino; Lineage 2: Tinaja and Pachón). To identify putative convergent losses of TFBS, we filtered for motifs that were present in at least 80% of all genomes in each surface population, and present in less than 20% of all genomes in each cave population (Supplementary Data 5).

## Enrichment analysis on candidate genes underlying repeated evolution

We conducted gene ontology (GO) enrichment analyses to ask whether the genes identified as putative candidates for convergent evolution of cave adaptation show overrepresentation of functional categories associated with cave-derived phenotypes. GO enrichment analysis may also reveal novel candidate phenotypes involved in cave adaptation. We individually analyzed each of the three sets of candidate genes (allele reuse: $n = 34$; locus reuse: $n = 3590$; overlapping sweeps: $n = 760$).

We first conducted an enrichment analysis with ShinyGO (http://bioinformatics.sdstate.edu/go74/; accessed 10/1/2022)[113] by providing the list of Ensembl gene IDs for candidate genes and comparing against the *Astyanax mexicanus* reference database (26,698 genes total), with a specified false discovery rate (FDR) of 0.1. We also conducted enrichment analysis with the GO Consortium Gene Ontology Enrichment Analysis tool (http://geneontology.org/; accessed 10/1/2022[114–116]) using genes that had associated gene symbols in the *Astyanax mexicanus* Ensembl v101 gtf (allele reuse, $n = 17$; locus reuse, $n = 2178$; overlapping sweeps, $n = 626$). For this analysis, Fisher's exact tests were performed to determine whether the number of genes associated with a given ontology were over- or under-represented in our set of candidate genes relative to the set of 25,698 genes in the reference database for zebrafish (*Danio rerio*).

As a control comparison, we also conducted the same enrichment analyses on the set of 172 genes showing signatures of a selective sweep in Mante and Rascón surface populations and neutral evolution in Pachón, Tinaja, and Molino cave populations. Our expectation was that these genes should not show a significant enrichment of gene ontologies or QTL associated with cave-derived phenotypes.

## The mechanism of repeated evolution across lineages

We used a coalescent modeling approach to test for patterns consistent with alternative modes of repeated evolution in our candidate genes. The Distinguishing Modes of Convergence (DMC) method developed by Lee & Coop[8] was implemented in the package rdmc[46] in R (v4.0.2). As this analysis is computationally intensive, we chose to focus on the candidate genes identified as evolving repeatedly via allele reuse across seven caves with AF-vapeR ($n = 34$) and using the overlapping sweeps approach in Pachón, Tinaja, and Molino cave populations ($n = 760$ genes) (see above for details). The list of 760 overlapping sweeps candidate genes also includes 150 of the 3590 genes identified by AF-vapeR as evolving repeatedly via locus reuse (i.e., frequency shifts in unique alleles within each lineage).

DMC models neutral evolution (no selection) and several alternative modes of repeated evolution, including migration, standing genetic variation, and independent de novo mutation, using population genetic data. A model of migration assumes an adaptive allele evolved in one cave population and then spread among other cave populations via gene flow. A model of standing genetic variation assumes that the adaptive allele was segregating at low frequencies in the ancestral surface population and then repeatedly increased in frequency in each cave population due to selection. Lastly, a model of de novo mutation assumes that the adaptive allele evolved independently in different cave populations via a novel mutation. Our expectation is that genes evolving repeatedly via allele reuse across populations from both cave lineages would be more likely to have support for standing genetic variation or migration spreading the same adaptive allele between lineages. Conversely, we expect de novo mutations might play a larger role in genes showing evidence of locus reuse.

For each of the candidate genes for repeated evolution, we provided DMC with allele frequencies from the three cave populations included in the overlapping sweeps analysis (two Lineage 2: Pachón and Tinaja; one Lineage 1: Molino) and three surface populations (one Lineage 2: Rascón; two Lineage 1: Río Choy and Mante) plus 10 kb upstream and downstream of each gene. Including a 10 kb buffer on either side of each gene helped to ensure that we could capture the decay in coancestry upstream and downstream from the selective sweep and to investigate whether the strongest signatures of selection tend to be focused in coding or noncoding regions. We specified Pachón, Tinaja, and Molino as the populations under selection for the loci analyzed. In addition to modeling the three main modes of repeated evolution in these three selected populations (i.e., migration from a single cave population into the other two, selection on an existing allele standing in each population, or independent mutations arising in all three populations), we implemented two mixed models where the two Lineage 2 populations were grouped together. These mixed models specified (1) migration spreading the adaptive allele between Pachón and Tinaja and an independent mutation in Molino and (2) selection from standing genetic variation targeting the same allele in Pachón and Tinaja and an independent mutation in Molino. To allow DMC to contrast neutral and selected allele frequency covariances among populations, we extracted all intergenic sites throughout the genome and thinned them to only include one site every 50 kb. This resulted in a set of 20,531 neutral allele frequencies.

The three main modes of repeated evolution are discriminated by DMC by examining variance in neutral allele frequencies and coancestry in selected populations around the putatively selected sites. The rate of decay of coancestry between and within selected populations differs depending on the underlying mode of repeated evolution. In the case of independent origins of the adaptive allele, we expect to see enhanced coancestry within selected populations and no increase in coancestry between selected populations, as selection is acting on unique mutations between populations. With selection from standing variation and migration, we expect to see increased coancestry within selected populations and also increased coancestry between selected populations, as selection is acting on the same variant across populations. While an independent mutation scenario is relatively easy to discern from an allele-reuse (standing or migration) scenario, distinguishing between standing and migration models depends on the time the allele was standing before the onset of selection and the number of migrants per generation (see below).

DMC calculates a composite likelihood for each alternative model at each site along a genomic region using a grid of parameter values. We specified an effective population size of 100,000 (which is close to the average effective population across cave and surface populations) and a recombination rate of 1.16 cM/Mb (estimated from the median recombination rate from ref. 109). We specified for DMC to calculate composite likelihoods at 50 evenly spaced sites across each region (i.e., encompassing the gene and 10 kb up and downstream) using all possible combinations of values specified for the selection coefficient, standing time (number of generations before the onset of selection and before migration from the source population occurred), migration rates, and frequency of the standing variant. See Supplementary Table 10 for specific parameter values used. The maximum composite likelihood for each model was identified and the composite likelihood for each alternative model of selection was compared to the neutral model for each of the 50 sites.

We also used DMC to infer the mode of repeated evolution in a "control" set of 172 genes identified as with diploS/HIC as having hard or soft sweeps in at least two surface populations (Mante and Rascón) and neutral evolution in Pachón, Tinaja, and Molino cavefish populations. All DMC parameters were the same as described above for the analysis of candidate genes for repeated evolution in cavefish, but the three surface populations (Rascón, Río Choy, and Mante) were specified as the populations under selection.

## Reporting summary
Further information on research design is available in the Nature Portfolio Reporting Summary linked to this article.

## Data availability
All sequencing data generated for this project is available under PRJNA558458. Accession numbers for each sample used in sequence analyses are provided in Supplementary Data 1. The surface fish *Astyanax mexicanus* genome assembly, Astyanax_mexicanus-2.0 (accession # GCF_000372685.2) is available on NCBI. The surface fish *Astyanax mexicanus* genome assembly, Astyanax_mexicanus-2.0 (accession # GCF_000372685.2) is available on NCBI. Ensembl's Biomart (v104) is available at https://www.ensembl.org/biomart/martview/. Dataset containing population genetic summary statistics for each gene in the genome is available at Dryad (https://doi.org/10.5061/dryad.3xsj3txmf)[117]. Source data are provided as a Source Data file. Source data are provided with this paper.

## Code availability
Custom code and scripts used in processing and analyzing the data associated with this manuscript can be accessed on GitHub (https://doi.org/10.5281/zenodo.7706730 [118]; https://doi.org/10.5281/zenodo.7706736 [119]).

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

## Acknowledgements

We thank the University of Minnesota Genomics Center for their guidance and performing the cDNA library preparations and Illumina HiSeq 2500 sequencing. The Minnesota Supercomputing Institute (MSI) at the University of Minnesota provided resources that contributed to the research results reported within this paper. We thank Ramses Miranda Gamboa, Carlos Pedraza Lara, Ulises Rivera-Arroyo, and María de los Angeles Verde-Ramírez for their support in during sample collection. We thank Josephine Paris, Silas Tittes, and Daniel Schrider for helpful discussion and guidance on data analysis. Funding was supported by NIH (1R01GM127872-01 to S.E.M., A.C.K., and N.R., R01DE025033 to J.B.G.) and NSF (IOS 165674 to A.C.K., IOS 1933076, IOS 2202359, and IOS 1923372 to J.K., S.E.M., and N.R., DEB 2147597 to J.K. and A.C.K.), and a US-Israel BSF award to A.C.K.

## Author contributions

R.L.M. and S.E.M. conceptually designed the project, conducted data analyses, and drafted and revised the manuscript. C.P.O.G. assisted with tissue sample collection for sequencing. J.B.G. and N.R. provided sequence data. E.J.R., C.P.O.G., J.B.G., A.D., J.W., A.C.K., J.E.K., and N.R. contributed to data analyses and interpretation and to writing the manuscript.

## Competing interests

The authors declare no competing interests.
