## [Peer Review File · Nature Communications]

Selection-driven trait loss in independently evolved cavefish populationsREVIEWER COMMENTS

Reviewer #1 (Remarks to the Author):

This manuscript uses the *Astyanax mexicanus* system of surface fish and multiple cavefish populations to explore the role of selection in constructive and regressive trait evolution. Cutting edge methods are used to scan sequenced genomes of multiple individuals in various surface and cave populations to provide new and important evidence resulting in the following conclusions: (1) present-day cavefish populations originated from two separate cave invasions of different surface fish ancestors, which happened at about the same time, thus providing the independent evolutionary trajectories to be studied in this paper, (2) regressive cave phenotypes, such as eye and pigment loss, evolved (at least in part) by selective sweeps, (3) repeated evolution of traits in the two cavefish lineages occurred nearly simultaneously by selection on standing genetic variation and by de novo mutations, and finally (4) genes evolving repeatedly in cave populations are longer, presumably because they are better targets for de novo mutations. The conclusions of this paper are in general robust, provide compelling explanations for previous presumptions in this field, and are important for future progress in this system. They also provide a paradigm for studies on other similar systems, including other cavefish lineages.

This reviewer has no major criticisms of this well done and exciting study. Below I provide some (minor) suggestions offered to increase clarity, to mention non-excluded possibilities, and to generally improve the story.

(1) The idea that cave constructive traits have been molded by selection has not been a historical area of conflict in this field. However, oppositely, the role of selection in regressive traits has been and is still actively debated. The authors provide strong evidence that regressive traits may also be under selection, which may be one of the most important findings of study. For this, they give one example, *oca2*, a pleiotropic gene under a QTL that controls pigmentation, which appears to show a signature of selection. It would enhance this conclusion if the authors provided evidence for selective signatures of a few other genes, particularly those under eye QTL or previously predicted to have roles in eye development. Some of these are *rx3* and *cbsa*, both under eye QTL, and *shha*, a pleiotropic gene thought to control many cavefish traits, including some regressive traits, but there are probably others. Adding information on eye genes would strengthen the conclusion about selection having a role in regressive, as well as, constructive evolution.

(2) This work does not exclude the possibility that some regressive cave traits may actually be a result of neutral mutation. The best evidence stems from QTL direction results for melanophore number. To be fair, this should be mentioned, perhaps somewhere in the Discussion. Most current cave biologists think that both neutral mutation and natural selection combine to drive cave trait evolution, and it is much more difficult to obtain strong empirical evidence needed to support or reject the neutral evolution theory.

(3) An important part of this manuscript is the finding of correlation between genes involved repeated evolution and QTL. In examining Figure S14, however, one sees many repeated reuse loci outside of (known) QTL, and the figure in general does not very well reflect the conclusions made in the Results. Many repeated genes are outside of known QTL and vice versa some QTL regions seem devoid of candidate genes for repeated evolution. To be fair, this should be mentioned in the Results. How do the authors interpret genes with repeated reuse outside the QTL? Are there many more QTL that remain to be discovered? This is one area in which the authors, who are otherwise very cautious in their conclusions, could venture some worthwhile speculation.

(4) The major conclusion that longer genes are more prone to de novo mutation is an obvious interpretation of the data. However, in light of another conclusion of the paper, that de novo mutation in non-coding rather than coding regions may be more important in the evolution of cavefish traits, the authors need to more precisely define what they mean by a gene in reference to the former conclusion. Do they mean only the coding region (as is stated on line 123 but nowhere else in the manuscript), or the coding region plus various noncoding regions, or something else? If non-coding regions (such as introns) are included in the definition then more opportunities may exist for mutation, and gene sizes could differ greatly but may have similar sized coding regions subject to mutations affecting translated proteins.

(5) It is very interesting that cavefish seem to have a higher de novo mutation rate than surface fish. On Line 618, aside from what is mentioned here, could another (unmentioned) quality

affecting mutation rate be the physical cave habitat itself? There is a theory that radon radiation, ubiquitous in the underground environment, could affect mutation rate.

(6) Line 71. "Caves have much lower light". Except for the entrance zone, these caves are characterized by no light at all.

(7) Typos. Line 557: add "to" between "needed" and "drive".

(8) Finally, this is a complex paper that was undoubtedly difficult to organize and write. However, some parts would greatly benefit from a tightening of the ideas and less redundancy. For example, lines 298-319.

Reviewer #2 (Remarks to the Author):

In this manuscript, the authors scan genome-wise convergent adaptive signals using 18 cave and eight surface populations of *Astyanax mexicanus* throughout the range in northeastern and central Mexico. The phylogenetic analysis led to two lineages, suggesting at least two independent evolved cavefish populations. In response to environmental pressures, cave populations have repeatedly evolved both regressive and constructive traits, which made *Astyanax mexicanus* an ideal model to study the molecular mechanisms of repeated evolution in response to similar environmental pressures. By using large-scale data and cutting-edge genomic methodologies, the authors found that 1) selection rather than genetic drift, has played a pivotal role in the evolution of adaptive traits, both regressive and constructive; 2) Selections on standing genetic variation and de novo mutations both contribute to repeated adaptation; and 3) Longer coding genes contained more repeated adaptive alleles. Overall, this is a well conducted and well-written study providing important information for repeatability evolution. Below I give my comments and a few questions for clarifications.

1. The whole analysis was genome-wise and lack of analysis for special region or genes, especially those drive trait loss. Could you describe the adaptive evolved trajectory for some instance please?
2. The supplementary tables in excel files are not matchup well. The table number should be clarified in each file or sheet, and figures shouldn't be included in the form.
3. L161-164: The author revealed the phylogenetic placement of Subterráneo cave as sister to Micos surface, and explain that the reasons for this result is the gene flow between a Micos region cave (Subterráneo cave) and surface fish from the Lineage 1 (Mante surface). However, why the Subterráneo population not clustered with Mante, but clustered with Micos surface population? Additionally, the Treemix graph indicates that there is gene flow between Subterráneo cave and Mante surface, but the ADMIXTURE bar plot does not shown admixture genetic structure between Subterráneo and Mante, can the authors give a reasonable explanation of the results?
4. L161: Label of Lineage 1 or 2 is different between Figure 1C and S1-S3.
5. L192-193: "This suggests that selection likely played a dominant role in trait evolution in cavefish." This conclusion is too far ahead here since the results before have no connection with any traits yet. The analysis only shows the genome of cave population is under selection compared to the surface ones.
6. L206-208: How about the enriched functional categories of genes in lineage-shared sweep windows? And how about the enriched GO for the genes that were selective sweep in surface populations but neutral evolution in cave populations?
7. L233: The time for surface stocks invaded caves was estimated be 160-190k generations ago here. However, in the corresponding figure legend (Figure 2A), the number is 150-250k generations. Which one is correct? In addition, why use the generations rather than years ago here? I think that the molecular dating should be performed to show a more intuitive result about when the cave populations originated, and the geological background should be included in discussions.
8. L248-253: Why chose the Pachon, Tinaja and Molino populations in this analysis? The approach required three or more populations, if other cave populations, or all cave populations were analyzed here, whether the results will be different a lot?
9. L255-265: AF-vapeR quantify and compare parallel genotype change based on allele frequency changes between population pairs. The identified windows are not subjected to selection necessarily. And the author used only single surface population in this analysis which may lead to too many windows. It is better to run with different surface population, and keep shared

windows/loci.

10. L268: "Figure 3E" should be Figure 3B?

11. L271: "Figure 3E" should be Figure 3F?

12. L292-294: "We suspect that this discordance may be because AF-vapeR is more powerful methods..." This can explain why the AF-vapeR detected more reuse genes, but not for the discordance between AF-vapeR and diploS/HIC methods. If it is because of one approach is more powerful, it should be that one result contains most of the other. By the way, the authors declared "many genes were excluded from the overlapping sweeps method if there was any evidence of sweep in surface populations." Was the analysis for selective sweep in surface populations performed? Are there any genes that exist in reused locus or allele but absent in overlapping sweeps here?

13. L298-300: How many populations were used in QTL studies? Are they the same as this work?

14. L315-319: Does it mean that there are more than half (54%) of genes evolving repeatedly across surface lineages overlapped with QTL? What's the functions for those genes?

15. L359: How many "allele reuse" or "locus reuse" are fixed in cavefish population? How many are within coding region and are them positively selected?

16. L377-382: The 33 and 760 genes were identified by different methods with seven and three cave populations, respectively. Whether the population number affected the DMC analysis for standing gene variation or de novo mutation?

17. L454: "... be upstream in 97 genes and downstream in 78 genes" How to define the locus upstream or downstream in genes? For example, 1kb, 2kb or 10kb?

18. L562-573: If the change of non-coding regions are responsible for the regressed traits, for example blind, in cave populations, some genes related to eye development might also show relaxing selection in the coding region, because there's no selective pressure for eyes and no need to keep the coding region stable?

19. L617-618: As the authors discussed that the mutation rates of the genome may be higher in cavefish populations compared to surface fish populations, however, the mean nucleotide diversity (π) in cave populations of lineage 1 and 2 (L1C & L2C) was obviously lower than the surface populations (L1S & L2S) (Figure S17). What might be the possible connections between the higher mutation rate and lower π value of the cavefish population?

20. Figure 2: Was the Figure 2A exactly the same with the Figure S12? Figure 2F, the Yerbaniz population seems to show younger selective sweeps compared to other cave populations. What's the possible reason for that?

21. Figure S5: The Treemix graph indicate gene flow between the surface population Mante and Nicara (Figure S5), but the Nicara was not mentioned in the manuscript, so what the Nicara belongs to? Lineage 1 or 2? surface population or cave population?

22. Figure S17 and S18 were not mentioned in the main text.

REVIEWER COMMENTS

Reviewer #1 (Remarks to the Author):

This manuscript uses the *Astyanax mexicanus* system of surface fish and multiple cavefish populations to explore the role of selection in constructive and regressive trait evolution. Cutting edge methods are used to scan sequenced genomes of multiple individuals in various surface and cave populations to provide new and important evidence resulting in the following conclusions: (1) present-day cavefish populations originated from two separate cave invasions of different surface fish ancestors, which happened at about the same time, thus providing the independent evolutionary trajectories to be studied in this paper, (2) regressive cave phenotypes, such as eye and pigment loss, evolved (at least in part) by selective sweeps, (3) repeated evolution of traits in the two cavefish lineages occurred nearly simultaneously by selection on standing genetic variation and by de novo mutations, and finally (4) genes evolving repeatedly in cave populations are longer, presumably because they are better targets for de novo mutations. The conclusions of this paper are in general robust, provide compelling explanations for previous presumptions in this field, and are important for future progress in this system. They also provide a paradigm for studies on other similar systems, including other cavefish lineages.

This reviewer has no major criticisms of this well done and exciting study. Below I provide some (minor) suggestions offered to increase clarity, to mention non-excluded possibilities, and to generally improve the story.

We thank the reviewer for their enthusiasm for the work and for their helpful suggestions and insight. We have addressed each comment below.

1. The idea that cave constructive traits have been molded by selection has not been a historical area of conflict in this field. However, oppositely, the role of selection in regressive traits has been and is still actively debated. The authors provide strong evidence that regressive traits may also be under selection, which may be one of the most important findings of study. For this, they give one example, *oca2*, a pleiotropic gene under a QTL that controls pigmentation, which appears to show a signature of selection. It would enhance this conclusion if the authors provided evidence for selective signatures of a few other genes, particularly those under eye QTL or previously predicted to have roles in eye development. Some of these are *rx3* and *cbsa*, both under eye QTL, and *shha*, a pleiotropic gene thought to control many cavefish traits, including some regressive traits, but there are probably others. Adding information on eye genes would strengthen the conclusion about selection having a role in regressive, as well as, constructive evolution.

We thank the reviewer for this suggestion and have added text at lines 327-354 in the manuscript adding information on the association between eye size QTL and genes under selection in caves.

Table S4 provides details on the diploS/HIC selection scan results for each gene in the surface fish genome annotation. Below we've included details on the genes mentioned above by the reviewer, followed by enrichment analyses on candidate gene sets.

- *rx3* is linked to a region under selection in cave populations and experiencing a hard sweep (a sweep was not detected within the gene itself but it occurs in close proximity to a genomic region experiencing a hard sweep)
- *cbsa* has evidence of a hard sweep in Pachón, Tinaja, and Molino, neutral evolution in Mante and Choy (Lineage 1 surface) and soft sweeps in Peroles and Rascón (Lineage 2 surface). Also identified as under repeated selection across cavefish lineages via locus reuse in the AFvapeR analysis (see Table S14).
- *shha* – We note that there are two genes corresponding to the gene symbol “shh” in the surface fish genome annotation, but ENSAMXG00000034481 is the ortholog to *shha* in zebrafish based on synteny. This gene has evidence of hard sweeps (from the diploS/HIC analysis) in Tinaja and Yerbaniz caves, as well as the Rascón surface population (Table S4), and was identified as under repeated selection across cavefish lineages via locus reuse in the AFvapeR analysis (see Table S14).

We found evidence that genes associated with previously identified eye size QTL are enriched (1) in genes under selection within individual cave populations and (2) in genes that are the repeated target of selection across lineages.

We found that 6,223 out of 26,698 genes total in the surface fish genome annotation (23%) are associated with previously identified eye size QTL, 5,977 of which had sufficient coverage for the diploS/HIC selection analysis (See Table S4).

Compared to the entire set of genes in the surface fish genome annotation, those identified as being under selection within any of the seven cave population analyzed (Table S6, Fig. S8) were enriched for eye size QTL (3,272 out of 12,389 genes under selection only in cave populations; Fisher's Exact Test, $P < 0.0000001$). Please see the new tab “EYE_QTL_GENES” added to Table S13 for more detail on this enrichment analysis.

Compared to the entire set of genes in the surface fish genome annotation, candidate gene sets for repeated selection were enriched for eye size QTL. Please see the new tab “EYE_QTL_GENES” added to Table S13 for more detail on these enrichment analyses.

- In the overlapping selective sweeps candidate gene set (genes with selective sweeps in Pachón, Molino, and Tinaja, neutral evolution in surface populations), 203 out of 760 (27%) of genes were associated with eye size QTL. This is a significantly higher proportion of genes than expected by chance (Fisher's Exact Test, $P < 0.01$).

- In the AFvapeR locus reuse candidate gene set (genes with evidence of parallel/repeated selection across seven cave populations), we also saw enrichment of genes associated with eye size QTL. Specifically, 35% of locus reuse genes were associated with eye size QTL, significantly higher than what would be expected by chance (Fisher's Exact Tests, $P < 0.000001$).
- No significant enrichment of eye size QTL genes was found in the AFvapeR allele reuse candidate gene set, but we note that this analysis was likely underpowered given the small number of genes identified with evidence of repeated selection via allele reuse ($n=34$ total).

2. This work does not exclude the possibility that some regressive cave traits may actually be a result of neutral mutation. The best evidence stems from QTL direction results for melanophore number. To be fair, this should be mentioned, perhaps somewhere in the Discussion. Most current cave biologists think that both neutral mutation and natural selection combine to drive cave trait evolution, and it is much more difficult to obtain strong empirical evidence needed to support or reject the neutral evolution theory.

We agree and have added text to the Discussion at line 634 noting that our findings indicating that regressive traits are under selection in caves does exclude the very real impacts of drift that can contribute to the evolution of these traits and that it is more of a challenge to investigate the role neutral mutations in regressive trait evolution:

“We note that while our results suggest a role for selection in regressive trait evolution, we absolutely expect that neutral evolution (and specifically genetic drift) also contributes to regressive evolution...”

3. An important part of this manuscript is the finding of correlation between genes involved repeated evolution and QTL. In examining Figure S14, however, one sees many repeated reuse loci outside of (known) QTL, and the figure in general does not very well reflect the conclusions made in the Results. Many repeated genes are outside of known QTL and vice versa some QTL regions seem devoid of candidate genes for repeated evolution. To be fair, this should be mentioned in the Results. How do the authors interpret genes with repeated reuse outside the QTL? Are there many more QTL that remain to be discovered? This is one area in which the authors, who are otherwise very cautious in their conclusions, could venture some worthwhile speculation.

This is a good point and ventures into the power (but also deceit) of QTL studies (to borrow from Beavis, 1994). Past QTL studies inevitably missed QTL due to being underpowered (often conducted with fewer than 300 F2 individuals) and the effect sizes of the QTL that have been discovered are likely overestimated due to the Beavis effect. Further, there may be a different genetic basis of a trait

depending on the environment the QTL was mapped in or the amount of drift that the laboratory lines experienced prior to being crossed. Generally speaking, very little biological meaning can be ascribed to the absence of a QTL.

Likewise, QTL studies were conducted mostly in Pachón, so we might not necessarily expect a high amount of overlap between QTL and candidate for repeated evolution (and we note that we found that only 30% of genes under selection in cave populations show overlap between both lineages).

The candidate genes for repeated evolution may be associated with cave adapted traits that have not been analyzed in previous QTL studies (e.g., metabolism, response to hypoxia) which have mainly focused on traits that are easy to score in the lab, such as tooth count, eye size, melanophore count, etc.; also see response to Reviewer Comment #21 below).

4. The major conclusion that longer genes are more prone to de novo mutation is an obvious interpretation of the data. However, in light of another conclusion of the paper, that de novo mutation in non-coding rather than coding regions may be more important in the evolution of cavefish traits, the authors need to more precisely define what they mean by a gene in reference to the former conclusion. Do they mean only the coding region (as is stated on line 123 but nowhere else in the manuscript), or the coding region plus various noncoding regions, or something else? If non-coding regions (such as introns) are included in the definition then more opportunities may exist for mutation, and gene sizes could differ greatly but may have similar sized coding regions subject to mutations affecting translated proteins.

Both coding region and transcript length were analyzed. In addition to the text referred to at line 123 (line 147 in the revised version of the text), these details were provided in the last section of the Results, titled “Genes evolving repeatedly across cave lineages have a greater mutational opportunity” (line 498 in the current revised version). We have added text at line 502 to clarify that by coding sequence length we are referring to CDS with exons excluded:

“...our analysis revealed that the set of 760 overlapping sweeps genes had, on average, significantly longer coding sequence length (CDS with exons excluded), more exons, and more predicted transcript isoforms compared to the reference database of all genes in the *A. mexicanus* genome (Figure 4B).”

5. It is very interesting that cavefish seem to have a higher de novo mutation rate than surface fish. On Line 618, aside from what is mentioned here, could another (unmentioned) quality affecting mutation rate be the physical cave habitat itself? There is a theory that radon radiation, ubiquitous in the underground environment, could affect mutation rate.

We thank the reviewer for their suggestion. This is an interesting hypothesis and we have added text to the Discussion mentioning the possibility that unique qualities of the cave environment could also affect mutation rate (now at line 712).

6. Line 71. "Caves have much lower light". Except for the entrance zone, these caves are characterized by no light at all.

We have corrected the text at line 89 to say that caves have no light.

7. Typos. Line 557: add "to" between "needed" and "drive".

This typo has been corrected. (Now at line 629).

8. Finally, this is a complex paper that was undoubtedly difficult to organize and write. However, some parts would greatly benefit from a tightening of the ideas and less redundancy. For example, lines 298-319.

We thank the reviewer and have made an effort to streamline the text throughout and have revised the noted section (which now begins at line 332).

Reviewer #2 (Remarks to the Author):

In this manuscript, the authors scan genome-wise convergent adaptive signals using 18 cave and eight surface populations of *Astyanax mexicanus* throughout the range in northeastern and central Mexico. The phylogenetic analysis led to two lineages, suggesting at least two independent evolved cavefish populations. In response to environmental pressures, cave populations have repeatedly evolved both regressive and constructive traits, which made *Astyanax mexicanus* an ideal model to study the molecular mechanisms of traits repeated evolution in response to similar environmental pressures. By using large-scale data and cutting-edge genomic methodologies, the authors found that 1) selection rather than genetic drift, has played a pivotal role in the evolution of adaptive traits, both regressive and constructive; 2) Selections on standing genetic variation and de novo mutations both contribute to repeated adaptation; and 3) Longer coding genes contained more repeated adaptive alleles. Overall, this is a well conducted and well-written study providing important information for repeatability evolution. Below I give my comments and a few questions for clarifications.

We thank the reviewer for the thoughtful feedback. We have addressed each comment and question below.

9. The whole analysis was genome-wise and lack of analysis for special region or genes, especially those drive trait loss. Could you describe the adaptive evolved trajectory for some instance please?

Thank you for the comment, a database of population genetic statistics, selection metrics, and phenotypic annotations for each gene in the surface fish genome can be found in Table S4. We have also added text at lines 346 – 368 in response to Reviewer 1's comments that specifically explores the impact of selection on eye-associated genes.

10. The supplementary tables in excel files are not matchup well. The table number should be clarified in each file or sheet, and figures shouldn't be included in the form.

Thank you for bringing this to our attention, we have double checked these references and included the table number and description within excel sheets (rather than just in the title of the excel sheet and in the supplemental materials). There was a pie chart off the side in one of the supplementary tables (which we believe is what the reviewer is referring to here) that we had generated for our own initial visualization of the data and had forgotten to remove – it has now been deleted.

11. L161-164: The author revealed the phylogenetic placement of Subterráneo cave as sister to Micos surface, and explain that the reasons for this result is the gene flow between a Micos region cave (Subterráneo cave) and surface fish from the Lineage 1 (Mante surface). However, why the Subterráneo population not clustered with Mante, but clustered with Micos surface population?

Additionally, the Treemix graph indicates that there is gene flow between Subterráneo cave and Mante surface, but the ADMIXTURE bar plot does not shown admixture genetic structure between Subterráneo and Mante, can the authors give a reasonable explanation of the results?

We thank the reviewer for these points and the opportunity to clarify.

The Micos River floods into Subterráneo cave, likely resulting in the admixture observed in this cave population. Mante was used as a proxy for Micos surface fish in the genome-wide admixture analyses due to sample size (and our phylogenetic analyses indicated Mante and Micos are both Lineage 1 surface populations).

More detail can be found in the Supplemental Material. For example, in lines 145-148 of Supplemental Materials:

“Mante served as the parental surface population for the initial training stage of the HMM, as we only had a sample size of one for the local Micos surface

population and our phylogenetic analyses revealed that Mante surface fish are closely related to the Micos and Subterraneo populations.”

And the following text is found starting at line 91 of Supplemental Materials: “Mante was included as the Lineage 1 surface fish population in these [admixture] analyses rather than Micos due to sample size (Micos: n=1; Mante: n=10).”

As presented in the Supplementary Materials, the Treemix analyses, D-statistic analyses in Dsuite, and HMM-based ancestry inference with Loter all showed strong evidence of admixture between Lineage 1 Surface fish (with Mante used a proxy for Micos) and Subterraneo cavefish.

ADMIXTURE uses a maximum likelihood based approach to assign ancestry coefficients to a pre-assigned number of genetic groupings in the dataset. This method does not employ any hypothesis testing (in contrast to Dsuite), and a recent publication (Kong & Kubatko 2021; full citation provided below) found that when the relative genomic contributions of the either parental population is very uneven and deviates substantially from 50/50 (i.e., if minor parent ancestry is closer to 0, as is common with later generation hybrids) ADMIXTURE sometimes fails to identify hybrids. We found that hybridization has been ongoing in Subterraneo for an estimated mean \pm SE of $9,843 \pm 600$ generations (with 1 generation/year assumed) with a mean major parent (surface) ancestry proportion of 0.84 and a mean minor parent (cave) ancestry proportion of 0.16. Thus, the asymmetry of parental populations’ contribution to the hybrid genomes may explain why ADMIXTURE did not pick up on the signal of admixture. To clarify, we have added text referencing this at line 183 of the Supplemental Materials:

“ADMIXTURE was also not evident in Subterraneo despite other approaches supporting admixture in this population (see below for further details). This discrepancy may be due to pronounced asymmetry in the relative genomic contributions from either parental population to the hybrid genomes (see below), as ADMIXTURE has been shown to fail to identify hybrids under these circumstances (Kong & Kubatko 2021).”

Kong, S., & Kubatko, L. S. (2021). Comparative performance of popular methods for hybrid detection using genomic data. *Systematic Biology*, 70(5), 891-907.

12. L161: Label of Lineage 1 or 2 is different between Figure 1C and S1-S3.

Thank you for the careful attention to detail and for finding this error. Labelling has been corrected in Figure S1 and S2.

13. L192-193: “This suggests that selection likely played a dominant role in trait evolution in cavefish.” This conclusion is too far ahead here since the results before

have no connection with any traits yet. The analysis only shows the genome of cave population is under selection compared to the surface ones.

Thank you for this comment, and we agree that this was a bit of an overreach. This line has been removed in the revised version of the manuscript. We appreciate the opportunity to correct this.

14. L206-208: How about the enriched functional categories of genes in lineage-shared sweep windows? And how about the enriched GO for the genes that were selective sweep in surface populations but neutral evolution in cave populations?

These are excellent points. We refer the reviewer to line 238 in the revised text, “GO terms that were consistently enriched in genes under selection in all seven cave populations but not enriched in genes under selection in either surface population included organ morphogenesis, circulatory and epithelium development, and cranial skeletal system development.”

GO terms enriched in the set of genes that had sweeps in surface populations and not in caves are provided in Table S7 and are also summarized at line 396 in the text:

“...the set of 172 genes under selection in surface populations and with neutral evolution in cave populations did not exhibit enrichment for known cave-adaptive phenotypes (Tables S7, S15). Instead, we observed a significant enrichment of ontologies related to broad functional categories, including regulation of gene expression, transcription, and meiosis. Overall, these analyses support the conclusion that cave-derived traits, even regressive ones, are shaped extensively by natural selection on a subset of genes that are the repeated target of selection in both lineages”

15. L233: The time for surface stocks invaded caves was estimated be 160-190k generations ago here. However, in the corresponding figure legend (Figure 2A), the number is 150-250k generations. Which one is correct? In addition, why use the generations rather than years ago here? I think that the molecular dating should be performed to show a more intuitive result about when the cave populations originated, and the geological background should be included in discussions.

We appreciate the reviewer’s comments, and note that dating analysis and discussion of geological background was performed in our previous paper Herman et al. 2018.

As noted in the caption for Figure 2, 150-250k generations before present was highlighted in the figure because this spans “...the range of previous demographic model-based median estimates for split times between cave and surface populations from each lineage from (Herman et al. 2018)).”

Following other recent work in the *Astyanax* system (e.g., Herman et al. 2018 Mol Ecol), we used a generation time of 1 year. This is stated in the Supplemental Results under the section “*Inferring Recent Demographic History*”, but for clarity we have also added this information to the main text at lines 257 and 977.

16. L248-253: Why chose the Pachon, Tinaja and Molino populations in this analysis? The approach required three or more populations, if other cave populations, or all cave populations were analyzed here, whether the results will be different a lot?

We appreciate the opportunity to clarify.

As noted on lines 1034-1036 in our revised manuscript, these populations had the highest sequence coverage and sample size and are the focus of most previous work in this system. Furthermore, while Pachón and Tinaja caves are both in the El Abra region (Lineage 2 cave populations), they are at opposite ends of the cave system with little evidence of gene flow between them. It has also been suggested previously (and potentially supported by the phylogenomic results we present here) that Pachón may represent a distinct evolutionary lineage of cavefish, but we have yet to identify the ancestral surface population. Thus, by including these three populations, we were able to test for signatures of repeated evolution within and between lineages.

Requiring a gene to have a sweep in at least 4+ cave populations and neutral evolution in surface populations (a prerequisite for analysis in DMC) would have been too restrictive for and led to a small number of genes that could be included in the analysis (<100). However, the AF-vapeR analysis was more inclusive and used seven caves total (three Lineage 1, four Lineage 2; see line 292 of the main text).

Lastly, we note that the coalescent modeling approach implemented by DMC is computationally intensive and expanding the analysis to include allele frequencies from additional cave populations would not have been feasible.

17. L255-265: AF-vapeR quantify and compare parallel genotype change based on allele frequency changes between population pairs. The identified windows are not subjected to selection necessarily. And the author used only single surface population in this analysis which may lead to too many windows. It is better to run with different surface population, and keep shared windows/loci.

The reviewer is 100% correct that we are assuming that highly parallel allele frequency change is an indicator of selection. We appreciate the opportunity to clarify, and have added that caveat on lines 994-996 and we have changed instances of “selection” to “allele frequency change.”

We want to clarify that we used two surface populations not one surface population, as mentioned in Table S13 and lines 993 in the revised manuscript. For this analysis, we used two surface populations (one from Lineage 1 and one from Lineage 2) with the highest sample size and coverage (see Fig. 3D, Line 180, Tables S1 & S3). Lineage 1 cavefish are derived from Lineage 1 surface ancestors, and Lineage 2 cavefish are derived from Lineage 2 surface ancestors. Thus, we identified regions of the genome where allele frequencies in cavefish populations are repeatedly moving in the same direction away from their surface fish ancestors. AF-vapeR was particularly useful because it identified loci where the two lineages experienced selection in the same direction (allele reuse) and loci where the two lineages both experienced selection but via unique genetic trajectories (locus reuse).

18. L268: “Figure 3E” should be Figure 3B?

Thank you for this careful attention to detail, this has been corrected.

19. L271: “Figure 3E” should be Figure 3F?

Thank you again for the attention to detail, this has been corrected.

20. L292-294: “We suspect that this discordance may be because AF-vapeR is more powerful methods...” This can explain why the AF-vapeR detected more reuse genes, but not for the discordance between AF-vapeR and diploS/HIC methods. If it is because of one approach is more powerful, it should be that one result contains most of the other.

We appreciate the opportunity to clarify.

We identified 3,615 unique genes within the locus and/or allele reuse categories with AF-vapeR. 3,563 of these genes had sufficient coverage for diploS/HIC results in at least one of the seven cave populations examined, and 100% of these genes had evidence of a selective sweep and/or linkage to a sweep.

Table S14 contains diploS/HIC results (i.e., evidence of a sweep or neutral evolution) for each gene in the “locus reuse” and “allele reuse” candidate gene categories for the cave populations. 84% of these genes (3,006 total) had evidence of a selective sweep in at least one cave population. A total of 1,924 genes (54%) had both a sweep in a cave population and neutral evolution in the corresponding surface population. Thus, there is substantial overlap between the diploS/HIC and AF-vapeR results. We have revised the text in question to provide further detail on Lines 325-330:

“Notably, all of the candidate genes identified by AF-vapeR were classified as having a sweep or being linked to a sweep by diploS/HIC (Table S14). We suspect that the discordance between the overlapping sweeps and AF-vapeR candidate gene sets may be because AF-vapeR considers phylogenetic context, and because many genes were excluded from the overlapping sweeps method if there was any evidence of sweeps in surface populations (a prerequisite imposed by the DMC analysis).”

We also note that the overlapping sweeps and AF-vapeR analysis imposed unique filtering approaches. Distinguishing Modes of Convergence (DMC) requires at least three populations under selection and three populations not experiencing selection to generate and compare models of selection and infer the underlying mechanism of repeated evolution at a given locus of interest. Thus, we only included genes where there were sweeps in Pachón, Tinaja, and Molino cave populations and neutral evolution in Mante, Choy, and Rascón surface populations (i.e., the “overlapping sweeps” gene set). Expanding to include additional cave populations would have dramatically reduced the number of genes included in the analysis (<30 genes had overlapping sweeps classified by diploS/HIC in at least 4 caves and neutral evolution in multiple surface populations; see Table S4). For the AF-vapeR analysis, we only focused on candidate genes where all seven cave populations showed signs of parallel/repeated selection (allele frequencies moving away from surface populations), so this analysis was also quite conservative.

By the way, the authors declared “many genes were excluded from the overlapping sweeps method if there was any evidence of sweep in surface populations.” Was the analysis for selective sweep in surface populations performed?

Yes, please see lines 396 - 408 and 483 - 495 in the text and Table S7.

Are there any genes that exist in reused locus or allele but absent in overlapping sweeps here?

Yes, 610 genes total. Please see Figure 3B.

21. L298-300: How many populations were used in QTL studies? Are they the same as this work?

This is an excellent point that was touched on by Reviewer 1, as well. We have now added a clarifier on Line 336-337.

Most previous QTL studies in this system have been limited to one cave population (almost always Pachón; see below). Kowalko et al. 2013 PNAS

included Pachón and Tinaja, and Kowalko et al. 2013 Current Biology only included Tinaja.

Previous QTL studies in *A. mexicanus*, focal traits, and population(s) included:
Protas et al. 2007 Current Biology – Eye and pigmentation regression, jaw size, tooth number, numbers of taste buds – Pachón
Protas et al. 2008 Evolution & Development – Eye size, melanophore count, relative condition, weight loss, tooth count, peduncle depth, fin placement, anal fin rays, suborbital bone width, rib count, length, chemical sense – Pachón
O’Quin et al. 2012 PLoS ONE – Retinal degeneration – Pachón
Yoshizawa et al. 2012 BMC Biology – Vibration attraction behavior, eye regression – Pachón
Yoshizawa et al. 2015 BMC Biology – sleep loss and prey-seeking behavior – Pachón
Kowlako et al. 2013 PNAS – feeding posture – Pachón and Tinaja
Kowalko et al. 2013 Current Biology – schooling behavior – Tinaja

22. L315-319: Does it mean that there are more than half (54%) of genes evolving repeatedly across surface lineages overlapped with QTL? What’s the functions for those genes?

We appreciate the opportunity to clarify. Less than 42% (72 out of 172) of genes evolving repeatedly across surface lineages overlapped with QTL regions (see Table S7 for details). As noted in the text at line 377, this does not deviate from what would be expected by chance ($X_1^2 = 1.51$, $P = 0.22$).

23. L359: How many “allele reuse” or “locus reuse” are fixed in cavefish population?

To allow readers full access to the data, we have provided extensive supplementary material.

Population genetic statistics for each gene are provided in Tables S4 and S14.

**Molino vs. Mante: 929 genes have a fixed difference (Table S14, column AE)
Pachón vs. Rascón: 336 genes have a fixed difference (Table S14, column AL)
Tinaja vs. Rascón: 142 genes have a fixed difference (Table S14, column AS)**

How many are within coding region and are them positively selected?

We were able to infer the specific site under selection in the genes that were run through DMC (which includes the allele reuse candidate from AF-vapeR). Of these, 62% were predicted to fall within the gene, but none were within exons. Please see Table S13, Column S.

100% of the AF-vapeR candidate genes for repeated evolution have evidence of being directly under selection or linked to a region under selection per the diploS/HIC analysis. See response to Comment #20 above for further detail.

24. L377-382: The 33 and 760 genes were identified by different methods with seven and three cave populations, respectively. Whether the population number affected the DMC analysis for standing gene variation or de novo mutation?

We thank the reviewer for the opportunity to clarify and direct them to the paragraph beginning at line 1138 of the main text for details. DMC was run using the same three cave and three surface populations for both sets of candidate genes.

25. L454: "... be upstream in 97 genes and downstream in 78 genes" How to define the locus upstream or downstream in genes? For example, 1kb, 2kb or 10kb?

As noted in Table S13 and in the text at lines 1141-1142, we included a +/-10 kb buffer region around each gene in the DMC analysis "to ensure that we could capture the decay in coancestry upstream and downstream from the selective sweep and to investigate whether the strongest signatures of selection tend to be focused in coding or noncoding regions".

We thank the reviewer for the opportunity to clarify, and we have also added text at line 525 in the Results to add further clarification.

26. L562-573: If the change of non-coding regions are responsible for the regressed traits, for example blind, in cave populations, some genes related to eye development might also show relaxing selection in the coding region, because there's no selective pressure for eyes and no need to keep the coding region stable?

The reviewer brings up an important possibility. We note that we found signatures of positive selection in genes associated with regressive traits, even when the site under selection was predicted to be in a non-coding region (see Table S13).

In response to a similar comment by Reviewer 1, we have now added the following text on lines 634-636:

"We note that while our results suggest a role for selection in regressive trait evolution, we absolutely expect that neutral evolution (and specifically genetic drift) also contributes to regressive evolution, and a recent study found little evidence of positive selection in *Astyanax* cavefish using divergence-based approaches limited to coding regions in one cave population (Pachón)⁶⁶."

27. L617-618: As the authors discussed that the mutation rates of the genome may be higher in cavefish populations compared to surface fish populations, however, the mean nucleotide diversity (π) in cave populations of lineage 1 and 2 (L1C & L2C) was obviously lower than the surface populations (L1S & L2S) (Figure S17). What might be the possible connections between the higher mutation rate and lower π value of the cavefish population?

It is possible to have a lower mean nucleotide diversity and higher mutation rate relative to the surface fish as these two metrics are not necessarily dependent on one another. The cave populations have low genome-wide nucleotide diversity due to the historical bottleneck event coinciding with initial cave invasion and lower effective population sizes of cave relative to surface populations (expected $\pi = 4N_e \times \text{mutation rate}$). Our investigation into the mode of repeated evolution (de novo mutation, gene flow, or standing variation) only focused on 794 genes with overlapping sweeps or evidence of repeated selection from AF-vapeR and neighboring regions (± 10 kb), whereas the mean π presented in Figure S17 was calculated in windows across the entire genome. Importantly, strong positive selection reduces nucleotide diversity (π) relative to what would be expected under neutrality. If an allele for a given gene is fixed in a population, π would be 0 for that gene.

28. Figure 2: Was the Figure 2A exactly the same with the Figure S12? Figure 2F, the Yerbaniz population seems to show younger selective sweeps compared to other cave populations. What's the possible reason for that?

Yes, Figures 2A and S12 are the same stairway plot and were included in each figure for ease of cross reference with the GEVA sweep age estimates.

The point about Yerbaniz is a very good one. We had previously discussed in older iterations of the manuscript but removed to save space in the submitted version.

Previous work has provided evidence that Yerbaniz cave experiences ongoing gene flow with the local surface fish population (Panaram & Borowsky 2005), which we hypothesize to be the cause of the younger estimated sweep times on average compared to the other cave populations. We have now added this to the main text at lines 265-268.

Panaram, K., & Borowsky, R. (2005). Gene flow and genetic variability in cave and surface populations of the Mexican tetra, *Astyanax mexicanus* (Teleostei: Characidae). *Copeia*, 2005(2), 409-416.

29. Figure S5: The Treemix graph indicate gene flow between the surface population Mante and Nicara (Figure S5), but the Nicara was not mentioned in the manuscript, so

what the Nicara belongs to? Lineage 1 or 2? surface population or cave population?

Thank you for the careful attention to detail. Nicara is short for *Astyanax nicaraguensis*, an outgroup congener from Nicaragua (please see line 764 of the main text and line 92 of the supplemental material). It is a surface fish. This has been further clarified in the legend for Figure S5.

30. Figure S17 and S18 were not mentioned in the main text.

We thank the reviewer for catching this. Figures S17 and S18 are now mentioned in the main text at line 808.

REVIEWERS' COMMENTS

Reviewer #1 (Remarks to the Author):

This manuscript has now been carefully revised and is now ready to be considered for publication. I am particularly satisfied to see clarification of selective sweeps in eye genes and QTL is the discussion.

Reviewer #2 (Remarks to the Author):

The authors have done a very good job in revising the manuscript. I am content and thankful for the careful responses and all my main points have been adequately addressed. I think this work will be a very valuable contribution to Nature Communications.